# Gendered and Racial Injustices in American Food Systems and Cultures

**Sally Kitch** [1,*], **Joan McGregor** [2], **G. Mauricio Mejía** [3] , **Sara El-Sayed** [4] , **Christy Spackman** [5] and **Juliann Vitullo** [6]

1   School of Social Transformations, Arizona State University, Tempe, AZ 85287-6403, USA
2   School of Philosophy, Historical, and Religious Studies, Arizona State University, Tempe, AZ 85281, USA; j.mcgregor@asu.edu
3   The Design School, Arizona State University, Tempe, AZ 85287-6403, USA; mauricio.mejia@asu.edu
4   School of Sustainability, Arizona State University, Tempe, AZ 875502, USA; sara.elsayed@asu.edu
5   School for the Future of Innovation in Society, Arizona State University, Tempe, AZ 875603, USA; christy.spackman@asu.edu
6   School of International Letters and Cultures, Arizona State University, Tempe, AZ 85287-0202, USA; juliann.vitullo@asu.edu
*   Correspondence: Sally.Kitch@Asu.edu

**Abstract:** Multiple factors create food injustices in the United States. They occur in different societal sectors and traverse multiple scales, from the constrained choices of the industrialized food system to legal and corporate structures that replicate entrenched racial and gender inequalities, to cultural expectations around food preparation and consumption. Such injustices further harm already disadvantaged groups, especially women and racial minorities, while also exacerbating environmental deterioration. This article consists of five sections that employ complementary approaches in the humanities, design studies, and science and technology studies. The authors explore cases that represent structural injustices in the current American food system, including: the racialized and gendered effects of food systems and cultures on both men and women; the misguided and deterritorialized global branding of the Mediterranean Diet as a universal ideal; the role of food safety regulations around microbes in reinforcing racialized food injustices; and the benefits of considering the American food system and all of its parts as designed artifacts that can be redesigned. The article concludes by discussing how achieving food justice can simultaneously promote sustainable food production and consumption practices—a process that, like the article itself, invites scholars and practitioners to actively design our food system in ways that empower different stakeholders and emphasize the importance of collaboration and interconnection.

**Keywords:** food systems; race; justice; food safety; diet; design

## 1. Introduction

Food injustices in the United States (U.S.) have multiple causes, at multiple scales. These injustices further harm already disadvantaged groups, especially racial minorities and women (Holt-Gimenez 2011; Odoms-Young and Bruce 2018). Such injustices are evident in the exploitation of underpaid and non-unionized immigrant farm and factory workers (Pulido 1996; Brown and Getz 2011). They are also reflected in historical patterns of discrimination that restrict food supplies in minority communities (Penniman and Washington 2018), in public policies that impose uniform nutritional standards on millions of people without regard for cultural values or individual needs, and in the decimation of Native food systems in Indigenous communities (Whyte 2015). In addition, the American food system's enormous carbon footprint—including its heavy reliance on fossil fuels and extravagant use of polluting pesticides and fertilizers—contributes to the social inequities exacerbated by climate change (Aleksandrowicz et al. 2016).

The industrialization and commodification of food in the U.S. is a key perpetrator of food injustice (Patel 2007; Gottliev and Joshi 2010). Such commodification undermines food's significance as a cultural, nutritional, creative, emotional, spiritual, and deeply personal aspect of people's lives. Policies and practices that situate eaters as consumers make eaters into passive selectors of available products rather than active partners in food production and food system design (Alkon and Mares 2012; Alkon 2014). Instead of increasing choice, as manufacturers claim, corporate, profit-based food economies actually reduce choice. Production of (often nutritionally inferior) industrial foods contributes to environmental degradation, worker exploitation, constrains the food supply, and devalues ethnic and Indigenous foodways (Nestle 2002; Whyte 2015).

The corporate food economy's perpetuation of gender and racial inequities found in wider society is another source of food injustice in the U.S. For example, gender and racial stereotypes render the low-level and poorly paid labor of women and minorities invisible, even though it is essential to commercial food preparation and production. In addition, prevailing gender norms frequently disadvantage women in the production and profit sectors of the food system. Racialized histories, stereotypes, and assumptions have reduced the availability of fresh foods in many communities. Meanwhile, industrialized food systems treat people like "embodied landfills", unevenly exposing Black and Brown bodies to "sugar ecologies" that decrease community health (Hatch et al. 2019, p. 603). Food justice activists themselves, who seek to create alternatives to the food insecurity of the industrialized agri-food system, may also back down from the challenging and time-consuming discussions of race and gender and the collective traumas and racialized inequities that alternative systems most need to address (Cadieux and Slocum 2015; Slocum and Cadieux 2015).

Food injustices have material as well as cultural consequences. The prevalence of White landowners (who own 98% of all U.S. farmland) and male farmers suggests systemic discrimination against women and minorities in agriculture (Horst and Marion 2019, p. 36). These historic injustices combine with economic and structural changes to undermine possibilities for more diverse food production practices (Leslie et al. 2019). Similarly, women in the kitchen are categorized as cooks, while men dominate the more valued, vaunted, and remunerated role of chef (Harris and Giuffre 2015). Minority communities that lack access to nutritious and/or culturally preferred foods also face race-based health disparities that become inscribed in the body in the form of diabetes, hypertension, and heart disease. Poverty and racial discrimination can exacerbate these challenges. Gender and racial discrimination also contribute to the incidence of eating disorders among women of all races, with especially harsh consequences for women of color (Julier 2019, p. 468; Thompson 2019). By the same token, market forces and popular narratives promote certain foods and diets that increase profits but do not account for the cultural contexts or practices.

This article explores such food injustices in five separately authored sections, paying particular attention to the racialized and gendered effects of the American agri-food system and the culture around it on both men and women; the misguided and de-territorialized global branding of the Mediterranean Diet as a universal ideal; the role of food safety regulations around microbes in reinforcing racialized food injustices; and the benefits of considering the American food system and all of its parts as designed artifacts that can be redesigned. We come from different disciplines and cultures, have had different experiences within the American food system, and even learned the English language in different ways. We thus retain our distinctive voices rather than attempting to create a homogenized tone.

The article concludes by discussing how design principles can guide actions to promote both food justice and sustainable food production and consumption practices. That process, like the article itself, requires us to redesign our food system in ways that empower different stakeholders and emphasize the importance of collaboration and interconnection.

## 2. Wrongs without Wrongdoers: Structural Food Injustices That Disadvantage Women and Minorities

*2.1. Dolores's Choices*

Dolores Bateman, a single mother of five children, lives in South Memphis, a predominately African American neighborhood without a supermarket accessible on foot to most of the area's residents (Jones et al. 2019). Dolores is a janitor at a local elementary school who must rely on public transit to and from her job and the grocery, a store which is a 45 min bus ride from her home. Many low-income children like those of Dolores get their meals covered by the federal free breakfast and lunch programs at school. While those school meals include key nutrients, they tend to be high in sugar, fat, and salt, and consist mostly of refined processed foods. This places them at risk for type-two diabetes (Hopkins and Gunther 2015). Lack of public amenities, parks or playgrounds due to redlining further exacerbates this risk.[1] A history of underinvestment means that low income minority neighborhoods have few commercial outlets (Badger 2017).

In short, Dolores's neighborhood is a "food desert" (Karpyn et al. 2019), although "food apartheid", a term for human-created systems of segregation that relegate certain groups but not others to food opulence, may better describe her neighborhood (Penniman and Washington 2018, p. 4; New York Law School Racial Justice Project 2012). These background conditions, most of which are not of Dolores's choosing, limit her food options. Entrenched social norms about gender roles mean women are usually the sole caregiver for children. These norms also track women into low-paid work. (Schieder and Gould 2016). This is not to say that Dolores has no agency in her food choices. She obviously wants a better life for herself and her children and tries to obtain healthy foods. But long work hours, constrained transit options, geographically limited access to healthy foods, and a low income, minimize her options. Social and environmental structures disadvantage her. Dolores is not in the same "choice" situation as middle-class suburbanites in neighboring cities. She must work harder to create a healthy life for her children.

Dolores's story reflects the unjust social structures in the U.S. that have perpetuated patterns of unequal food distribution (Elmes 2018). These patterns of unequal distribution are not necessarily enacted by individual wrongdoers. Rather, they result from "human-created" societal arrangements and, therefore, constitute social/political wrongs. "Social structures" include the background conditions, rules, policies, practices, and norms which govern individual actions, collective actions, and government actions. When social structures unfairly constrain or limit some people's opportunities and their capabilities vis à vis others in society, they create structural injustices. "Capabilities" refer to the set of valuable functioning—being well-nourished, having control over some property, being educated— that a person can effectively access. Mere freedom to consider these functionings is not sufficient; the key is effective freedom to choose among them and thereby to choose the life that one values (Nussbaum 2003).

*2.2. Wrongs with(out) Wrongdoers*

Most individual and governmental actions do not explicitly aim to limit opportunities or harm particular groups. These actions rather work within the norms of social structures, historically developed by and for privileged majority groups. Racist underpinnings of policies such as those that produce heavy concentrations of pollutants in minority communities, can render invisible how regulatory guidelines produce wrong outcomes (Downey 2005). Similarly, the underlying social structures framing food systems may remain invisible; they are products of systemic racism rather than the consequences of overt actions by current individual racists. These unjust social structures that appear inevitable, such as redlining

---

[1] Redlining was the discriminatory practice by lenders, backed by the federal government starting in the 1930s, whereby they would "redline" or flag communities or neighborhoods, mostly where minorities lived, and deny them mortgages due to their supposed higher risk of default. Richard Rothstein (2017) in *The Color of Law* (W.W. Norton and Company) details how the FHA subsidized builders creating suburbs with the requirement that no houses be sold to African-Americans. Even when redlining was explicitly outlawed in the 1960s, the underinvestment and lower property values continued.

and disinvestment in inner-city minority neighborhoods ([Rothstein 2017](#)), are products of human agency in the past. So too are gender expectations and apparently obvious norms around work that have historically limited women's ability to enter well-paying careers.

Though created and maintained by design and human agency, it remains difficult to identify any single racist or sexist actors in creating many of these unjust background social conditions. Neighborhoods may no longer be redlined, but redlining's legacy persists: limited access to housing loans has kept most BIPOC (Black, Indigenous, and people of color) families poorer than than White families by limiting residents' ability to build wealth in property ([Bloome 2014](#)). Other infrastructures, like corporate actions and government policies to subsidize commodity crops, have created an overabundance of cheap junk food swamping minority neighborhoods ([Franck et al. 2013](#)). Although not explicitly motivated by racism, that result has nevertheless harmed minorities and the poor ([Petersen et al. 2019](#)). Likewise, the continuation of those policies in agriculture and the food industry, which support corporate profits over human health, are not necessarily designed to harm minorities or the poor. Nevertheless, they have the same or similarly harmful impact on the poor, women, and minorities as explicitly racist or sexist policies do.

Despite the lack of intentional discrimination, then, there is a discriminatory outcome (known in law as a disparate impact). Injustice under such circumstances entails more than simply the fact that people suffer fates they do not deserve. Rather, it concerns how institutional rules and social interactions conspire to narrow the options many people have. As Iris Marion Young notes:

> Social structures do not constrain in the form of direct coercion of some individuals over others; they constrain more indirectly and cumulatively as blocking possibilities. Part of the difficulty of seeing structure, moreover, is that we do not experience particular institutions, particular material facts, or particular rules as themselves sources of constraint; the constraints occur through the joint action of individuals within institutions and given physical conditions as they affect our possibilities. ([Young 2012](#), p. 55)

Why consider these outcomes injustices as opposed to disadvantages that are not entitled to be redressed? Justice requires that individuals have the social conditions of freedoms to function as equal citizens. When there are structural barriers, particularly when those barriers are the result of governmental policies and other collective arrangements that restrict the social conditions for exerting freedom, those limitations constitute injustices.

Thus, if injustices are created by designed social structures—laws, policies, practices, and social norms—that largely result from human action and maintenance; structural injustices should be redressed by new or redesigned social structures ([Haslanger 2012](#)). Often, however, creating new structures further harms the disadvantaged. For example, responses to Hurricane Katrina—decisions about who received aid and what kind of aid— led to structures that narrowed opportunities for many victims, limiting their capacities to do and be various things ([Sen 1992](#), pp. 39–42). The social structures that create food apartheid function in a similar fashion. Living under food apartheid limits food choices for poor, female, and minority populations. In relation to others in society without those limitations, the victims of structural injustices are unequal and their capacities are diminished. The constrained capacities are not simply products of misfortune; they are the by-products of collective social arrangements.

### 2.3. Situating Responsibility

This account claims that the disadvantages and harms resulting from the food system are largely a result of the social structures which place complex and varied barriers in the way of individuals' option networks, closing off or not providing certain options. That does not mean that individuals have no agency. Individuals do make choices, but structural obstacles can undermine their sense of agency and social equality. Limited opportunity constrains the social conditions of freedom.

If social structures cause the injustices, then who is responsible for those injustices, for rectifying or compensating for them? Who can we blame for the food situation that Dolores finds herself in? Is it the grocery store chain that does not locate in her neighborhood? The executives of banking and insurance companies who perpetrated redlining are long gone. Dolores's education level, resulting in a low-paying job, and her role as a single parent are the consequence of an even more complex set of factors, including gender norms, unequal funding for school and child care where she lives, etc. There may be no clear individual bad actors in the background conditions that structure the relationships creating Dolores's unjust and unhealthy food situation. Yet, there is a need for accountability of decision-makers at all levels about how their choices have negative consequences for the low-income minorities and the planet. Solving the problems of injustice inherent in the food system requires changing those conditions, which can also be done by design. In Section 5, we offer some suggestions for a critical and systemic approach to designing food systems where decision-makers can actively understand the needs of disadvantaged groups and transform the social structures that harm them. Again, quoting Young: "promoting justice in social structure and their consequences implies restructuring institutions and relationships to prevent these threats to people's basic well-being" (Young 2012, p. 34). Focusing on individual wrongdoers perpetrating injustice leaves structural cases unaccounted for.

If we acknowledge that social structures oppress some groups in society, particularly women and minorities, and if we are committed to everyone having the social conditions of freedom in terms of basic capabilities for well-being (Anderson 1999, p. 316), then we are collectively responsible to change those social structures that restrict some people's social conditions for freedom. Although societies may not have to guarantee equality of resources, a just society is obligated to ensure that people's capacities for basic well-being are roughly the same. There is particular responsibility for redress if limitations on freedom result from social arrangements created and sustained by the political system, some of which were authorized by historical overt discrimination.

Acknowledging that harms perpetrated by the social system within which individuals act in relation to the food system identifies them as violations against the principle of providing people what they are owed. This means, among other things that all people deserve equal opportunities for developing their capabilities to live good lives through healthful eating (Nussbaum 2003). If their opportunities for well-being are constrained by collective social arrangements, then they do not have those capabilities—that is an injustice.

## 3. Women and Womanliness: Gendered Food Injustices in American Culture

### 3.1. The Hazards of Gendered Food Injustice

A young African American woman named Joselyn was told by her White grandmother from an early age that she was fat, even though she was not overweight. The grandmother teased Joselyn that she would never be as pretty as her lighter-skinned cousins. Soon Joselyn's Black father, whose business was booming, joined in admonishing her and her mother and sisters to lose weight. To him, the women's thinness signified the family's upward mobility. Even before puberty, Joselyn was put on diet pills and encouraged to downsize. At the same time, the father insisted on having large meals in his home and bought treats for his daughters. The confusion about food abundance and body size expectations, especially as related to class mobility, coupled with Joselyn's determination to be thin, since she could not change her skin color, led to a pathological cycle of dieting, compulsive eating, and bulimia from which she suffered well into adulthood.[2]

Joselyn's story about dysfunctional eating and the female body's connection with the social symbolism of food illustrates how talking about food means talking about gender. Food defines people, often in destructive ways, both as gendered individuals and as gendered members of families, races, social classes, communities, and nations (Weismantel 1992). Joselyn's is also an intersectional story, meaning that her social identi-

---

[2] This vignette is based on a true story recounted in Thompson (2019, pp. 186–87).

ties as a Black, middle-class, heterosexual, urban female function together. Biases, stereotypes, expectations, and cultural and material oppressions affecting her life are simultaneously racial, gendered, sexualized, and classed. Thus, Joselyn's eating disorder reflects the impact of trauma and intersectional sociocultural comorbidities, connected with gendered racial stereotyping and racism, on young Black women's self-esteem and body dysphoria, resulting in eating pathology (Hawthorne et al. 2017).

Not all American stories about gendered food injustice are linked to affluence. Indeed, the coronavirus pandemic has exposed and exacerbated many existing food injustices, as increasing numbers of people in the U.S., especially people of color, have tumbled into unemployment and food insecurity because of poor federal management of the pandemic's economic effects. Gender only compounds the negative impact of the increasingly unjust distribution of and access to food affected by race, geography (urban or rural), and economic status. That happens because social and cultural expectations typically imposed upon women exacerbate economic dislocation and marginalization suffered by communities. Thus, on top of race- and class-based food injustices that impact groups, such as non-unionized essential workers and Indigenous communities, women in those groups are often subject to food-related role expectations and prescriptions for their bodies and womanly identities. In short, specifically gendered collective background conditions, like those addressed in Section 2 of this article, exacerbate the injustices for women.

Drawing upon Joselyn's case and other examples and data, this section will analyze several of these gendered "super" oppressions through which women experience food injustice in two categories: (1) oppressions and deprivations in food cultures, systems, and industries; and (2) oppressions tied to racialized, heterosexualized, and classed ideas about womanliness in American culture.

### 3.2. Responsibility without Agency

Stereotypes, hierarchies, and discrimination in food cultures, systems, and industries constitute a major source of gendered food injustice in the U.S. At the root of this injustice is a contradiction. Women in many cultural and racial groups may be regarded as naturally inclined to provide food for others and, therefore, expected to do so as "good" women. At the same time, because of (hetero)sexist stereotypes and discrimination, many women in the U.S. continue to have limited control over food production systems. They are largely courted as consumers but not recognized as agents, even if they are the primary food providers for their families.

Thus, on the one hand, fulfilling the gendered expectation to feed her family can be a key aspect of an American woman's social value: she displays her womanliness and love for family through the food she serves (Figure 1) (Vester 2015, pp. 137–38). Advertisers often connect women's sex appeal, happy marriages, and healthy children with the act of feeding a family (Parkin 2006). This expectation has been especially strong for African American and immigrant women, who may be considered engineers of family connectedness through the meals they provide, despite costs to their own health (Reese 2018, p. 199).[3]

---

[3] Stereotypes of Black women as powerful food providers have generated accusations that they are natural castrators of black men (Avakian and Haber 2005, p. 24).

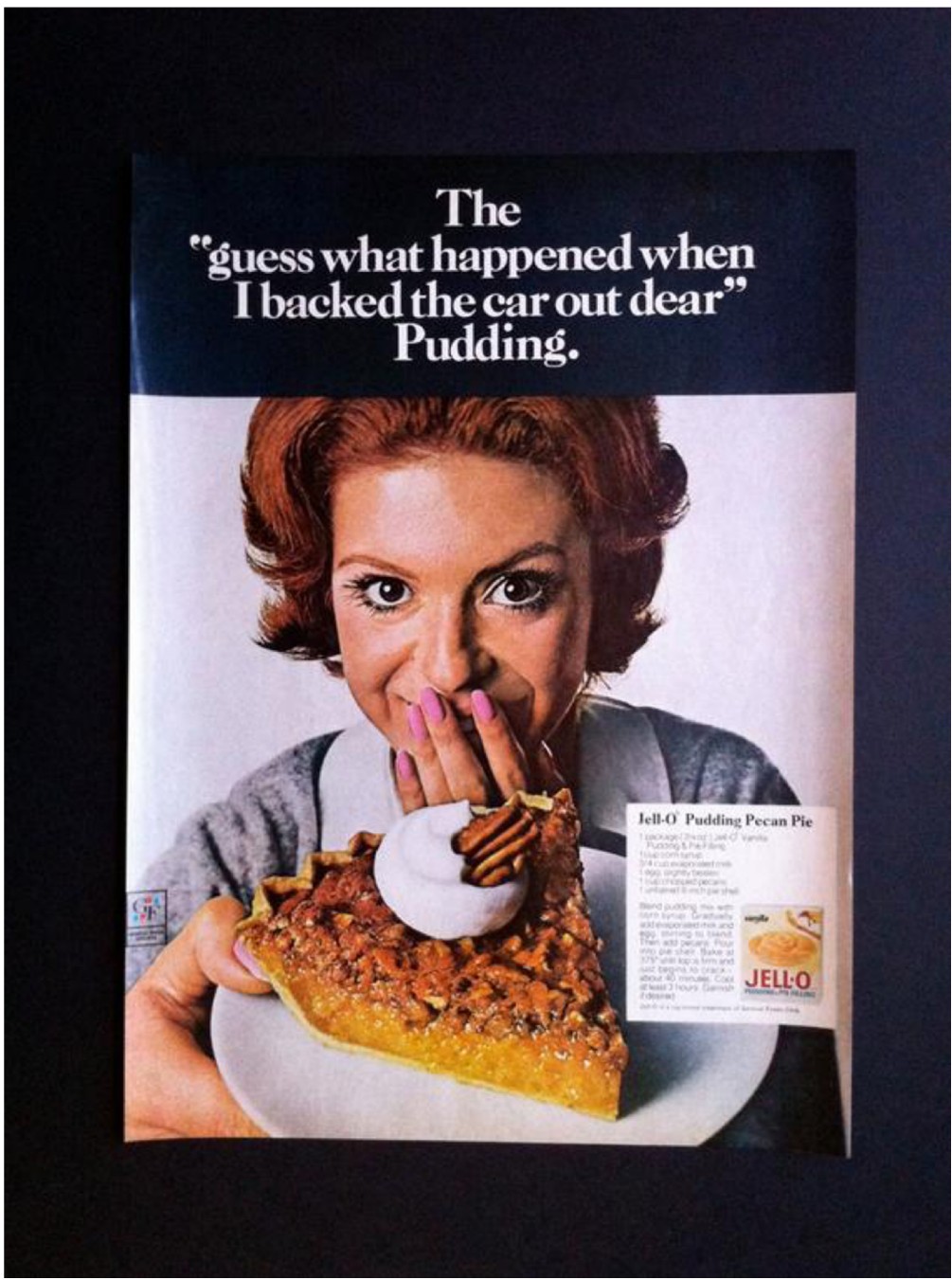

**Figure 1.** A 1963 Jello advertisement epitomizing stereotyped post-WWII associations of married White women's domestic and interpersonal powers with the provision of food[4].

For White heterosexual women even today, associations with food-giving model wives linger from post-World War II images, like the fictional Betty Crocker, who epitomized the fulfilled, perfect homemaker-cook. Some American women still struggle with that role expectation almost 50 years after Friedan's (1963) *The Feminine Mystique* highlighted the stultifying limitations of the housewife ideal. Contemporary "postfeminist" (usually White, heterosexual) women, who believe they are choosing to stay home, cook from scratch, and even raise their own chickens, may not realize the structural constraints on their choices or how their apparent choices can exacerbate gender inequities and patriarchy.

---

[4]  https://www.tias.com/stores/mspackratz/ (accessed on 2 February 2021). This image, like all others used in this essay, is in the public domain.

Thus, the contemporary young wife who, like her grandmother, bakes cookies to show her appreciation for her husband's role as a provider may not recognize that embracing her food-giver domestic role, however deliberately, can nevertheless reinforce the gender binaries that still produce gender injustice throughout society (Sharp 2018, p. 852).

Expectations of women as food givers also haunt contemporary lesbian women, who face their own food-related role stereotypes. Images of masculinized lesbian eaters, gnawing on rare steak with bared teeth, have historically been pitted against equally stereotyped images of the daintily nibbling "real" women associated with the heterosexual "food giver" role (Probyn 2018; Lindenmeyer 2006, p. 470). Lesbian writers, like Alice B. Toklas, have challenged dominant heterosexist interpretations of food-giving by embracing the role of cooking for and eating with a beloved same-sex partner (Vester 2015, pp. 166–69). More recently, queer theorists have contributed to a general understanding of the way that "emotionally charged foodstuffs become . . . shorthand for sexual identities or political standpoints". Instead of reductionist gender stereotypes, they assert the "complexities of eating/feeding relationships between lesbians", which can be symbolized by the varied eating preferences and cultural histories that shape a potluck meal (Lindenmeyer 2006, pp. 479, 481).

Alternative perspectives and food movements have not, however, eliminated even Lesbian mothers' assumed responsibility for the health and weight of their children in the U.S. (as in other countries). Indeed, the greater acceptance of lesbians as mothers has thrust them increasingly into "the system of meaning that envelops motherhood" in American culture (Lewin 2018, p. 190). The social construction of motherhood is even more connected than conventional wifehood to women's roles as food providers. This association further reinforces socially unjust gender binaries, despite differing sexual identities, obscuring the structural inequalities and stereotypes that shape what appear as individual choices.

Among those structural inequalities are limitations on access to healthy foods. Mothers who do not provide healthy foods for their children may be judged inadequate by those privileged with easy access to fresh produce. Yet, that disdain ignores the economic structures, housing policies, and racialized poverty that limit access to such foods for many Black, Latinx, or Indigenous women, as discussed in Section 2 of this article. The problem can be compounded by the resulting association of White, middle-class women with alternative food sources. Such racializing of farmers markets and organic food (Alkon 2012, pp. xi, 4) may alienate women of color from advertising campaigns promoting healthy products. It may also provoke censure for poor women seen buying expensive organic products instead of cheaper substitutes (Martin et al. 2019, pp. 185–87, 190). By the same token, racial associations with alternative food sources can obscure marginalized women's own alternative food practices, such as growing vegetables, participating in community gardens, creating mobile markets, and organizing ride-sharing to purchase fresh produce (Reese 2018, pp. 202–3).

On the other hand, and despite these stereotypical associations of food provision and preparation with heterosexual womanliness and motherhood, American women often lack agency in the operation and design of food systems. Male dominance in the high-status world of master chefs is one example of gendered obstacles to women's agency in the food industry. Even as numerous televised cooking shows featuring male chefs have brought more men into the domestic kitchen over the past 50 years, gender norms in professional cooking circles have changed little; women still hold a small fraction of head chef positions in the culinary industry—18.7 percent in the U.S.[5] The percentage of female chefs awarded Michelin Stars is about half that figure—9.2 percent. The disparity between the sexes in that prestigious and lucrative rating system underscores male privilege in the industry.[6] Women

---

[5] This figure is from the U.S. Department of Labor, 22 January 2021. https://www.bls.gov/cps/cpsaat11.htm (accessed on 2 February 2021). The percentage of female head chefs has increased from 4.7 percent in 2014 (Walkinshaw 2014).

[6] "Five Female Chefs with Michelin Stars: Get Inspired by Them!" *Artemis.* https://www.artiemhotels.com/en/blog/chef-female-michelin-star.html (accessed on 2 February 2021).

of color are a third less likely than White women to be accepted as chefs or authorities on food, thanks to their intersectional social disempowerment (Nettles-Barcelón et al. 2015).

Moreover, men's increased involvement in home kitchens, "often comes with a disavowal of feminized cooking". Cis-gender men tend to preside at weekend barbeques but participate less in the everyday routine of feeding a family. Thus, gender injustice persists—or possibly increases—as men invade the traditionally female kitchen domain and capture status and profit by connecting cooking with masculine prowess. Even more unjust are the bullying and sexual harassment women face in the macho cultures of commercial restaurants, where rates of sexual offenses are higher than in other employment sectors (Herkes and Redden 2017, p. 126).

Gender stereotypes also persist in farming, despite American women's recently increased control of agricultural land and capital, self-identification as farmers, participation in agriculture without a male partner, and growing influence in sustainable production practices (Sachs et al. 2016). Nevertheless, farming women, especially in the Mid-West U.S., are still ridiculed for considering themselves farmers. Some report that peers and consumers expect them to offer advice about canning vegetables or making jams, like stereotypical farm wives. One farmer explained that her wholesale clients would not talk to her. "They were always looking for Brad [her husband, who is not a farmer]". "There remains considerable public speculation about women's ability to farm . . . for the public and . . . farmers themselves" (Wright and Annes 2020, pp. 376–77).

Such obstacles limit women farmers' opportunities for financial independence and hamper their sense of empowerment from their non-traditional role. They often face an inhospitable climate among their male peers, as well as in financial programs and agricultural organizations. Because the masculine coding of farming remains strong, as the above examples suggest, women farmers who dare to infiltrate a masculine arena like an Agricultural Extension meeting "are often viewed as wives or cheerleaders to promote male interests" (Wright and Annes 2020, p. 379).

### 3.3. Women as Food

Another source of racialized and heterosexualized gendered food injustice is the cultural equation of food itself, rather than the role of food provider, with women's identities, social value, and sexiness. The advertising industry has for decades made that equation visible by assuming the male gaze—judging and commodifying women's bodies according to White heterosexual male standards of female value—and by objectifying women as consumable goods through associations of food with female bodily shape and sex appeal (Adams 2014; Ponterotto 2016; Cairns and Johnston 2015). Touting specific diets for women and men is part of that commodification, which further distorts people's relationship to the food they consume. Since the American food industry "creates three times as many food products as [our] society need[s]", it can justify spending "billions of dollars selling endless variety and constructing new needs" and tastes in food to create demand for its products (Julier 2019, p. 468). Gendering and sexualizing food is part of that sales-pitch (Figure 2).

Thus, food advertising, packaging, and appearance in movies and other media convey gender- and sexually coded messages that expose the vexed relationship between gender and power in the U.S. (Inness 2001; Contois 2020). For example, thin White women are typically associated with salads, diet foods, and small meals, which thus become signs of high-status femininity (Figure 3). Meanwhile, men of all races (and maleness itself) are associated with large, hearty, spicy meals and processed snacks, dubbed by some marketers as "dude food" (Contois 2020). Abundant meat is overwhelmingly coded as masculine and associated with power (Brissette 2017). Indeed, across societies, meat-eating conveys a message of male dominance, as eaters are thought to absorb a slaughtered animal's prowess. At the same time, meat's presence on the table dis-empowers women, who are the "absent signifier" implied in the masculine consumption of flesh. Women are thereby

devoured with the meat: "swallowed and . . . swallowers, consumers and . . . consumed" (Adams 2014, p. 241) (Figure 4).

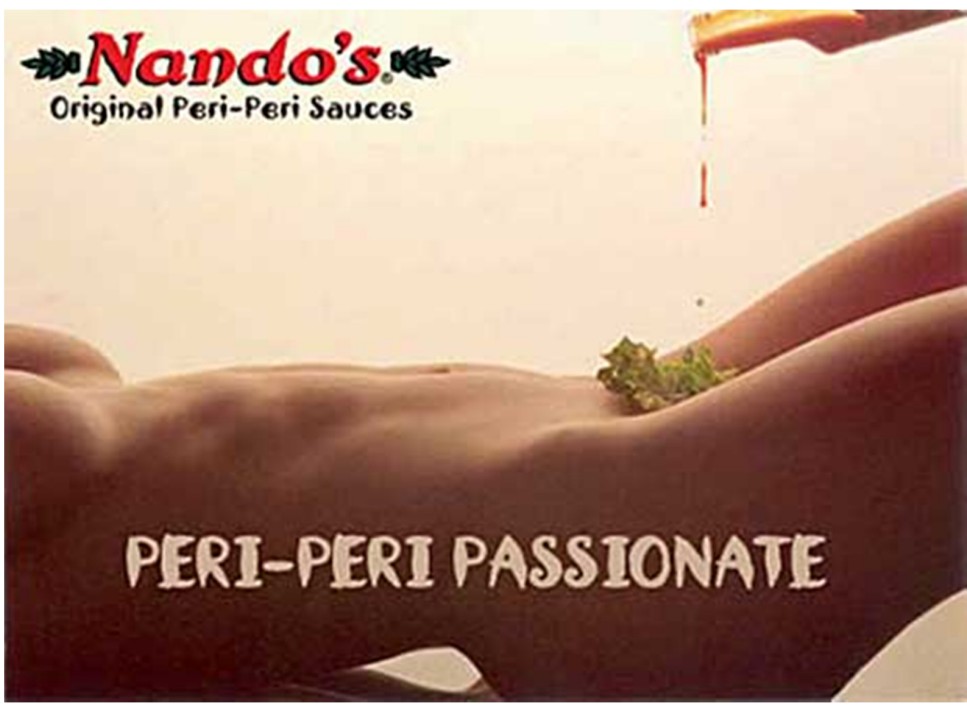

**Figure 2.** Women as literal food, Nando's advertisement, 2015. Chiowama—Beauty Blogger, http://chiowamabailey.blogspot.com/2011/02/my-top-5-fast-food-restaurants.html (accessed on 10 December 2020).

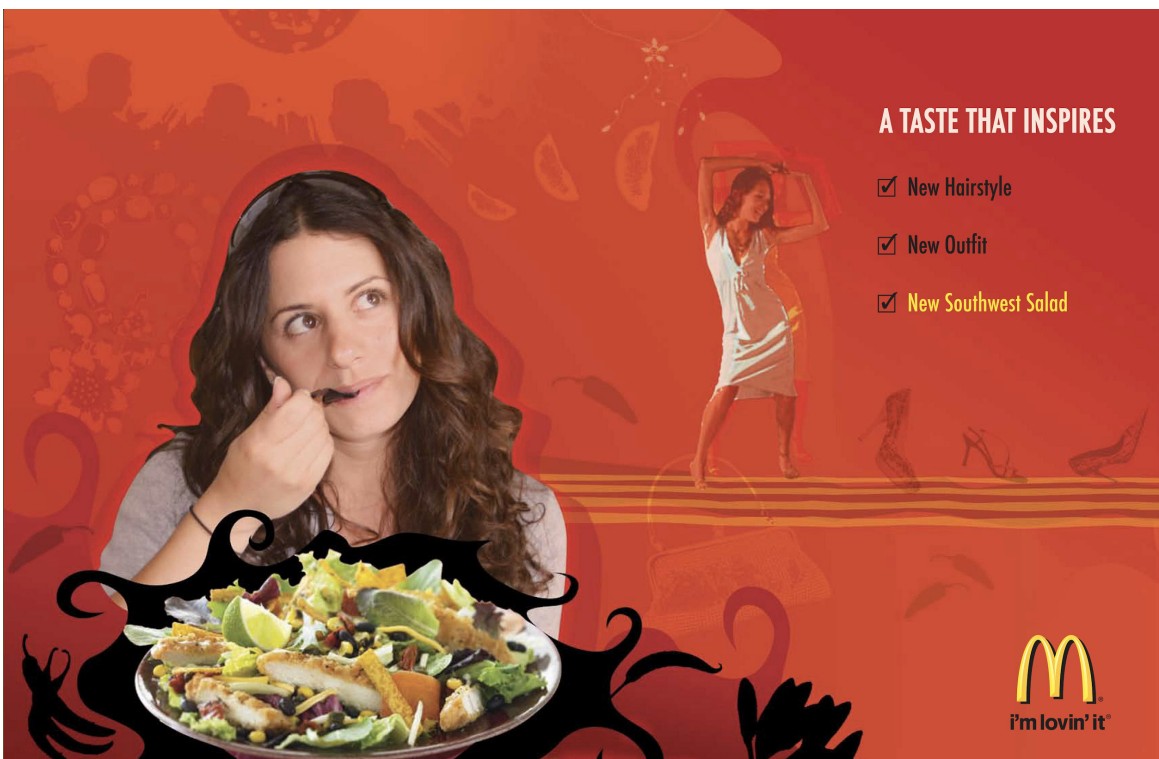

**Figure 3.** Body control as key to social value and status for women. MacDonald's advertisement, 2007. https://english1105.wordpress.com/2012/10/08/advertisement-thesis/ (accessed on 10 December 2020).

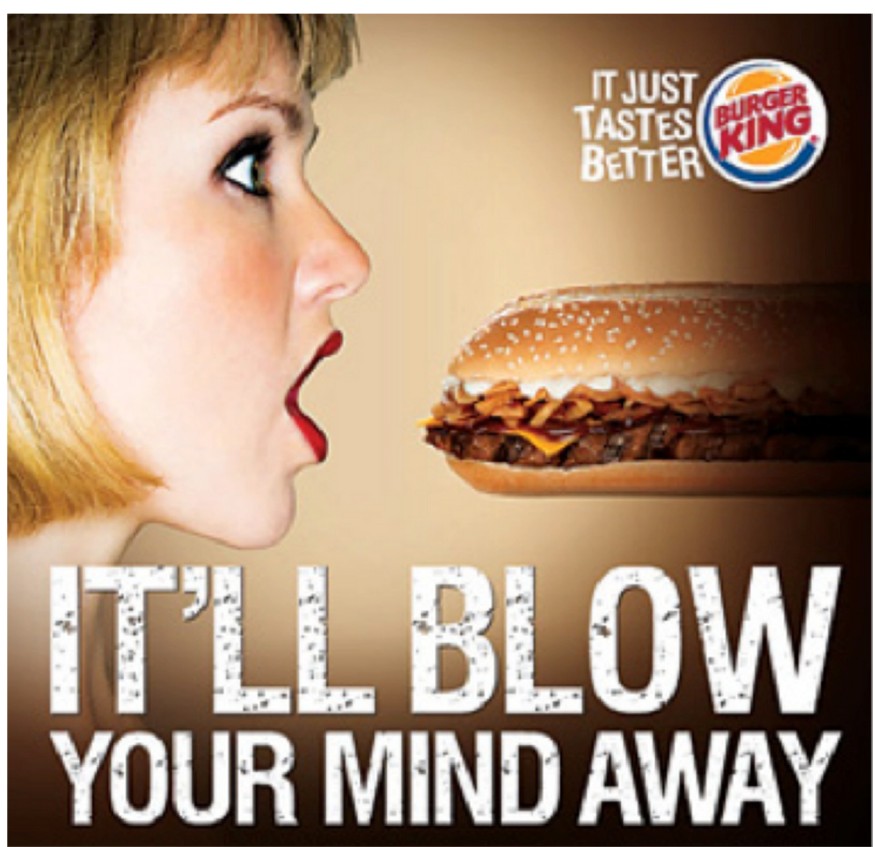

**Figure 4.** Women as heterosexualized, implicitly swallowed, swallowers of meat. Burger King advertisement, 2014. https://heyhowsitgoin.wordpress.com/2012/01/20/pictures-of-women-depicted-negatively/ (accessed on 10 December 2020).

These gender associations have material, sometimes life-threatening, consequences for both sexes. For women, the association of high-status (White) femaleness with low-calorie diets can produce body anxiety and eating disorders, as it also stokes male fantasies about female bodies (Jovanovski 2017).For men, meat-heavy, highly processed junk-food diets can engender numerous potentially fatal diseases, such as diabetes and heart disease. Moreover, masculine imagery associated with that diet makes some men reject healthier foods. To counter the latter hazard, both Weight Watchers and Nutrisystems launched multi-million-dollar campaigns in the early 2000s to attract men to diet programs that had historically been 90 percent female. Program advertising highlighted male-centered sports and celebrity images to convince men that dieting was not solely a feminine pursuit, although the dieting technologies for men and women were identical (Contois 2019, pp. 124–25). That the campaign worked moderately well demonstrates how vulnerable both men and women can be to gendered fantasies about food's relationship to their bodies' social value.

Joselyn's story further demonstrates that women of color can be as susceptible as White women to cultural messages about the desirability of controlling female bodies for social purposes. That means they are also susceptible to eating disorders, such as anorexia, bulimia, and compulsive overeating and dieting in seeking or being frustrated by alleged feminine ideals (Hawthorne et al. 2017; Thompson 2019). The thinness-as-status message reinforces heterosexism and racial stereotypes related to food cultures and "obscures the underlying structures of inequality that foster [individual] problems". At the same time, that message implicitly and explicitly demonizes fat female bodies, especially those of lesbians and women of color (Julier 2019, p. 468; Thompson 2019). These food-related stereotypes and biases can lead to further prejudice and social injustice (Figure 5).

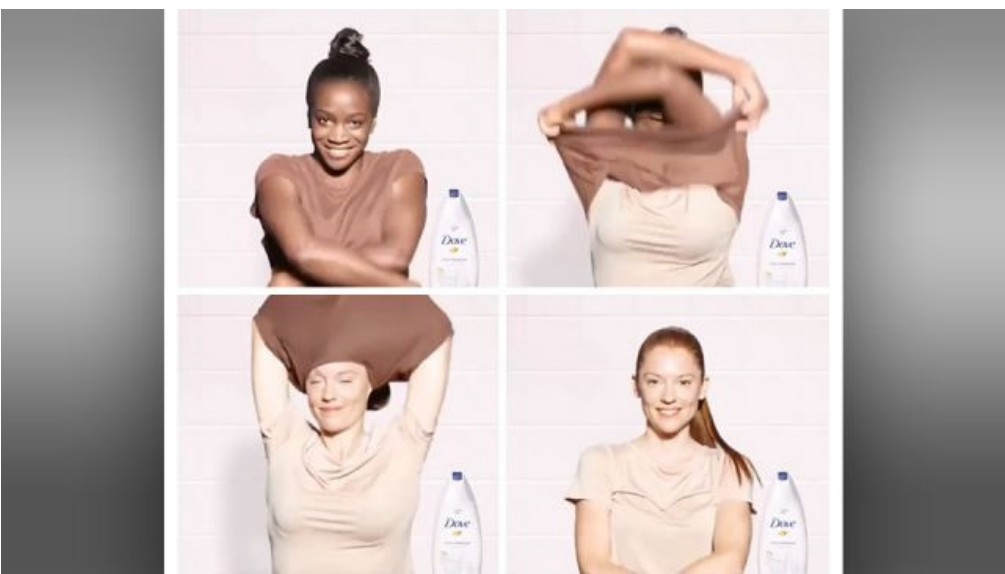

**Figure 5.** Racist context for food injustice. Social Media advertisement for Dove Body Wash, 2017. The ad caused a furor and was featured in numerous tweets, internet articles, posts, and podcasts, including: https://www.nydailynews.com/entertainment/tv/dove-model-racist-ad-not-victim-article-1.3556259; http://myspice.tv/dove-suffers-backlash-over-racist-ad/; https://www.socialsamosa.com/2017/10/dove-racist-ad-furore/; https://www.thealternativedaily.com/dove-apologizes-for-controversial-body-wash-ad/ (all accessed on 10 December 2020) Dove apologized for the racist implications of the ad, which a spokesperson said were unintentional. However, the company's lack of awareness is part of the problem of racism in the U.S. The Black model said she had no idea how her pictures would be used.

## 4. The "Ideal" Mediterranean Diet?: The Risks of Promoting De-Territorialized Foodways without Cultural Context

In the 1950's the now-famous physiologist from the U.S., Ancel Keys, "discovered" the Mediterranean diet (MD) during a research trip to Naples. Together with his wife Margaret Chaney Keys, he popularized that diet in their best-selling advice manual, *Eat Well and Stay Well* (Keys and Keys 1959). Ever since, tension has existed between Keys's place-based research and the call for this diet, based largely on plant foods, seafood, olive oil, and limited consumption of meat and dairy products, to be universally translated around the world (Anderson and Sparling 2015, pp. 165–67). Keys's focus on the nutritional advantages of the diet and the scientific authority substantiating those benefits obscures the historical and ideological reasons for the development of those cultural practices in certain Mediterranean communities, which were the result of subsistence food producers' creativity in the face of poverty, oppression, and the threat of malnutrition.

### 4.1. The De-Territorialization of the Mediterranean Diet

It is often remarked that Keys embodied the MD; he lived in Cilento, a region in Southern Italy, for over 30 years, joining the ranks of the community's famous centenarians. However, Keys' life in Cilento carried distinct economic, racial, and gender privileges that facilitated his own access to fresh food and good health. Historically a poor region, Southern Italy and its predominantly farmworker inhabitants have faced significant discrimination. In contrast to many of his neighbors, Keys could afford a gardener and a cook. His personal chef and housekeeper, Delia Morinelli, was an important knowledge source for local foods and recipes. These recipes were often passed down orally through generations of mothers and daughters (Moro 2014, p. 46). Yet, Morinelli has only recently received some recognition for her contributions to Keys's research. In recent interviews, Morinelli and other women of her generation and their daughters explained how women participated in vegetable and wheat production, milled their own flour, made their own

breads and pasta, learned to conserve important foods in their culture such as eggplant and anchovies, and transported fish from the seaside to the hilltop towns where families practiced terraced farming and produced olive oil (Granai del Mediterraneo 2013; Interview with Giuseppina and Rosetta by Elisabetta Moro n.d.). Keys obscured the knowledge and experiences of these women by translating their Cilentan traditions into a scientific paradigm.

Instead of focusing on the MD as an Indigenous set of food practices developed over centuries by peasants, a tradition in which women played a crucial role in production, Keys created his own mythological roots for his "discovery". From his adopted home in the seaside town of Pioppi he could see the ruins of the Ancient Greek city of Elia, the apparent location of one of the first schools of medicine in the West. In that setting, Keys saw himself and the other male scientists who joined him in Cilento as historical heirs to an authoritative tradition of Western knowledge about health. He combined the name of his home city in the U.S., Minneapolis, with that of the Ancient Greek city Elia to form the title of his Pioppi-based scientific community, Minnelea (Moro 2014, pp. 114–18). Although he noted that Minneapolis was a name derived in part from the Sioux word for water, Keys never considered that the Indigenous populations of the U.S. might have place-based food traditions that were equally worthy of study (Moro 2014, pp. 118–19). With the reduction of Pioppi's food system to a nutritional schema that could supposedly be applied to any person in the U.S. and beyond, especially wealthy American men suffering from high rates of heart disease, the MD lost its historical and geographic context and actual creation stories.

If Keys's scientific paradigm of the MD started the de-territorialization of Cilento's practices, the movement to have it recognized as one of the first food traditions considered a UNESCO Intangible Cultural Heritage of Humanity continued the process. Starting in the 1990s, experts from governmental and non-governmental organizations, scholars from universities, and business organizations like the International Olive Council successfully linked the scientific construct of the MD to the Med Label, which represented the interests of the agro-food market. The benefits to that market were obvious: "It is clear today that the olive oil lobby . . . made a visionary move with a durable effect" (Marques da Silva 2018, p. 580). In 2006 these same interests initiated a campaign to have the MD officially recognized as an Intangible Cultural Heritage by UNESCO, granted in 2010 (Marques da Silva 2018, p. 583; Nestle 1995; Nestle 2018, pp. 173–75). In this way, the UNESCO process of recognizing the importance of traditional foodways was appropriated to market certain products, especially olive oil, rather than used to address contemporary public health challenges and the racial and gender histories that shaped them.

### 4.2. The Dangers of the "Best" Dietary Model

Keys's notion of the MD as a universal solution for nutritionally based chronic conditions persists despite dramatic increase of those diseases during the decades-long voluminous record of academic studies and the diet's popular promotion. For instance, from 2018 to 2020, the *U.S. World and News Report* proclaimed the MD as the best diet of the year. That influential magazine's version of the MD illustrates "nutritionism", the perils of privileging quantifiable analyses of discrete nutrients while avoiding discussions of economic, social, or cultural perspectives (Scrinis 2015, pp. 259–60; Kimura et al. 2014, p. 37). Although Keys's model for the MD was a community of relationships in Cilento, in which people, land, water, animals, and plants shared a deep history, media descriptions emphasized individuals and weight loss. The Cilentan variety of the MD, developed through many generations as a form of collective knowledge and practices to deal with evolving challenges, including food scarcity, is now ironically used to promote food restriction, especially for women.

The 2019 Eat-Lancet Commission's report on Food in the Anthropocene also chose the MD, which it defines as "similar to the diet of Crete in the mid-20th century", as the primary reference model for the future's healthier and more sustainable diet. The commission chose the MD as representative of plant-based Indigenous food traditions throughout the globe because it is the "best studied" (Willett et al. 2019, p. 454). Although the report emphasizes

that other traditional diets also offer healthy eating patterns, including the consumption of less red meat, it does not question the failure by scholars and nutritionists to pay similar attention to non-European alternatives.

Continued promotion of the Mediterranean as the "best" dietary model brings up several concerns. This dietary advice is based on a model developed in a largely pre-industrialized society, when traditional practices, intergenerational relationships, and embodied learning created production and consumption habits that are impossible to replicate in contemporary urban, or even rural areas of the U.S. With the push to imitate food practices that focus on seasonality, fresh rather than processed food, and commensality, the burden of achieving those nutritional goals often falls on women, especially mothers (as discussed in Section 3), "without giving them resources to provide better food, such as flexible work hours, reductions in the gender wage gap, and changes to a welfare system that has pushed many women to low-wage jobs" (Kimura et al. 2014, p. 41). The responsibility to ensure that all children are food secure should belong to the whole society, yet a continuing focus on the nutritional requirements for individuals and the moralizing approach to the issue of childhood obesity evades larger questions about economic structures and policies. This makes it difficult for parents to dedicate time and other resources to feeding their families healthy foods.

Additionally, the MD model perpetuates a colonialist, racialized dietary hierarchy (Kimura et al. 2014, p. 38). The MD today is usually associated with White European identities (Spain, Italy, and Greece). Its "best" ranking is due to decades of scientific research that have ignored other Indigenous foodways. As a result, many BIPOC communities' traditions remain under-studied even though they also offer plant-based models, like certain Native American communities' emphasis on the three sisters (corn, squash, and beans). These traditional foodways encourage the consumption of more legumes and less meat, one of the goals of the Food in the Anthropocene report for a healthier, more sustainable world (Wall Kimmerer 2013, pp. 128–40; Willett et al. 2019, pp. 455–57). The promotion of the MD could, in fact, have the opposite effect of pushing immigrant families or vulnerable communities to question the healthfulness of their own traditional practices (Moffat 2020).

*4.3. Re-Territorializing Traditional Foodways*

In the Southwest U.S., recognizing and valorizing Native food traditions is particularly important. Diné storyteller, Sunny Dooley, notes that the pandemic has revealed the cultural and physical traumas perpetuated against Native populations that have left them vulnerable. Dooley explains that the violent colonization of Diné culture and territories produced a crisis in their land-based spiritual and food practices, which led to both emotional trauma and chronic physical illnesses, such as diabetes. "COVID is revealing what happens when you displace a people from their roots". She notes: "The truth is the disparity: of health, well-being and human value. And now that the truth has been revealed, what are we going to do about it"? (Dooley 2020). Dooley highlights the clear nutritional and environmental health inequities among Indigenous peoples. CDC data supports Dooley's concern, demonstrating that the incidence of COVID-19 among American Indian and Alaska Native people is 3.5 times more than for White people, especially among people under age 65 (Hatcher et al. 2020).

Maria Parra Cano of Phoenix, who identifies as a Xicana Indigena, asked a similar question to Dooley's in 2004, when both of her parents were diagnosed with diabetes. Parra Cano looked for answers within her family's own traditions for healing practices, as she had also developed nutritionally based chronic illnesses. She responded by attending culinary school. The French-based curriculum failed to meet her cultural and dietary needs. Therefore, she applied what she learned at culinary school to her knowledge of Mexico's Indigenous food traditions, gained from years of cooking alongside her mother (Zah 2020, p. 20). Using ingredients native to the Sonoran Desert, such as *nopales* (prickly pear pads) and *verdolagas* (purslane), Maria successfully healed her family and herself, enabling them

to stop taking medications for diabetes (Zah 2020, p. 22). Maria now disseminates her delicious, plant-based cuisine, together with cooking workshops and programs for children where she passes down her knowledge to the next generation. Maria's work demonstrates culturally meaningful and place-based food education, which recognizes the importance of diversity to human health (Hoover 2017, pp. 57–59).

The superficial translation of the MD, as promoted by agro-food interests, nutritionists, and the weight-loss industry, focuses on the universal prescription of olive oil consumption as the best source of lipids. It emphasizes individual consumer habits rather than a holistic approach rooted in geographical and cultural relationships. Clearly, the mass exportation of olive oil from Spain and Italy cannot on its own develop healthier food practices in the U.S., or even protect traditional practices in the Mediterranean territories producing the olive oil (Grosso and Galvano 2016, p. 16).

A better approach is a deep translation that re-territorializes foodways. Re-territorialization includes localized racial and gender histories in the study and practice of traditional foodways. It attends to Indigenous food cultures, often passed down by women, which have been rendered vulnerable by racial injustice. Seeing the MD as a cluster of distinct traditional foodways, which are also facing steep challenges by the industrialized, globalized food system, counters the "diet of the year" approach and opens the way to considering Indigenous traditions, both those where we live and those that are compatible with various cultural and biological ecologies, as guides toward developing healthier, more equitable, and more sustainably diverse relationships within food systems.

## 5. Food Contamination Standards That Can Foster or Reduce Food Injustice

### 5.1. Practicing Gotho in a New Land

"The concept of *gotho* is very important, it means impurity or contamination . . . I teach my kids no double dipping . . . taking a piece and eating it 'til it's done is fine, but taking a bite and putting back is unacceptable. If you take something, it's not returned", explained a Nepali mother and nurse practitioner in Mesa, Arizona, as she demonstrated traditional food safety practices for making *gundruk*. *Gundruk* is a fermented then sun-dried food made from sun-dried leaves of vegetables such as cauliflower or mustard greens (Figure 6). Even if Nepali communities are integrated within American society, they maintain a strong connection to traditional foods like *gundruk*. It is either produced in homes or brought back, often illegally, from trips to Nepal. U.S. customs control restricts the border crossing of many traditional foods, especially ones produced in collaboration with microbes through a fermentation process.

*Gundruk* production relies on tacit knowledge with its own norms and values passed down by generations (Fonte 2018). A light lactic acid fermentation produces *gundruk's* characteristic sour smell (Farag El sheikha 2018). Often used to create hearty soups and salads, *gundruk* is high in dietary fiber and carotene (Ranjan Swain et al. 2014). Traditionally produced in the Himalayas by women, *gundruk* usually circulates among friends and family; it is rarely bought from a stranger. These close networks depend on noses tuned to detect when the product is not properly produced through *gotho*, the traditional safety regulations that shape production to minimize the possibility of contamination. Producers recognize that traditional knowledge is part of the foods' safety protocols. Yet, women—and men—in the U.S. producing *gundruk* under the informal, extra-legal boundaries of *gotho* cannot bring their product to market without making major changes to the production method, changes that separate their foods from long tradition.

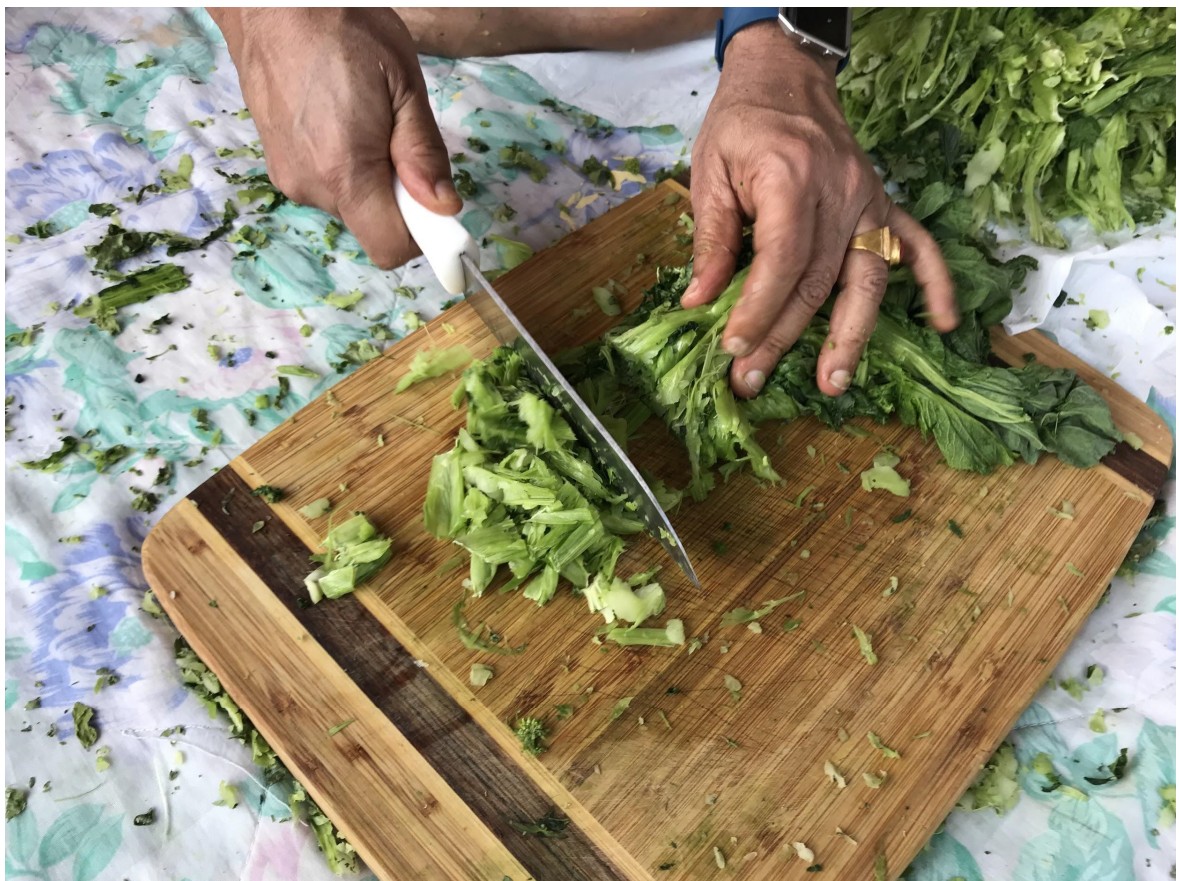

**Figure 6.** A Nepali, in Mesa, Arizona chopping the dried greens before placing them in jars for fermentation (Photo Credit: Nalini Chhetri, 2018).

People get sick from unsafe food. Figuring out that microbes in food make people sick was one of the central public health triumphs of the late nineteenth and early twentieth centuries. National and state governments responded by passing laws and regulations to minimize danger to the public. Informed by twentieth-century scientific insights, these legal structures have increasingly defined food production. Yet, food safety is not just science. It is a regulatory practice shaped by history and culture (DuPuis 2002). People have harnessed microbes for centuries as part of cultural practices to preserve foods like *lutefisk*, *nato* and *parmesan-reggiano* that link microbial action with traditional food cultures. Contemporary food safety regulations often sit uneasily alongside these traditional practices, in part because food safety regulations can easily slide into "regulatory phobias" that exclude some peoples and practices from marketplace access (Paxson 2019).

Food safety regulatory frameworks seek to order the world (Law and Mol 2008). We see this as a dual ordering. On one level, food safety regulations order physical materials to reduce the danger inherent in eating. On another level, food safety regulations order practices. These orderings tend towards binaries that pit humans against nature (Baur 2016; Merchant 1996) and promote a zero-tolerance policy; that is, zero tolerance for certain forms of microbial life (Wilson and Worosz 2014). However, in reaching that conclusion food safety regulations have stripped out cultural norms from the process of food production, replacing things like wooden shelves and vats with easily sanitized steel. As eaters with no interest in dying from our dinners, we see a lot of good in this ordering. However, we note that ordering practices also order people. In contemporary food safety frameworks, there exists little space for materials and practices outside of non-binary constructions, such as clean v. dirty, pure v. contaminated. This, we posit, unequally impacts women and minorities' ability to bring their traditional foods to market—foods

that oftentimes have significant health and cultural benefits but are produced in ways and through methods foreign to contemporary food safety structures. We are concerned that the increasing stringency of national and transnational food safety regulatory structures are producing regulatory phobias that disproportionately circumscribe traditional food production practices and the cultures they stem from.

In order to examine how language about contamination contributes to a larger atmosphere of regulatory phobia, we draw on newspaper articles around food safety in the state of Arizona as well as interviews with members of the Arizona-based Nepalese community who produce *gundruk*. Regulatory tensions around *gundruk* include its domestic locations of production and approaches that rely on tacit knowledge, resist using stainless steel, and expose foodstuff to the elements. As *gundruk* has a legacy as a safe and sanitary foodstuff, these practices, while producing tensions with regulators, strongly suggest that food safety can exist without the excessive technological and fiscal intervention that often keeps traditional foods from market. Perhaps a more expansive lexicon for speaking about food safety—one that extends beyond contamination—could help mitigate some fears about traditionally produced foods that have emerged from culturally situated regulatory phobias. Changing how legislators and regulators manage microbial life first requires changing how they talk and think about microbes, moving beyond contamination and seeing opportunities for more culturally inclusive regulations.

### 5.2. Talking about Food Safety

The language currently used to describe food safety reflects a reductionist and often binary lens. This "food safety lexicon" is not codified. Rather, the language used to describe food safety creates an informal lexicon that reflects current regulatory and social understandings of food safety. To capture this lexicon, we conducted a literature review of newspaper articles published in Arizona from 2011 to 2019 on Food Safety News[7], the *Phoenix Business Journal*, *ProQuest Magazine*, and NewsBank. Arizona has been especially impacted in the last five years by food-borne illnesses traced to foods produced in the state, making it an ideal locale for considering these questions. We conducted a general Google search of news articles for the following keywords: Food, Contamination, Outbreaks, Microbes, Fermentation, *Salmonella*, *E.coli*, *Saccharomyces*, Probiotic. The majority of articles found reflected negative impressions of microbes. Of the 73 articles identified in our searches, 51 focused on negative aspects of microbes and 40 focused on food-borne illnesses. The majority of articles discussed different types of outbreaks. Primary among them was the 2018 discovery of *E. coli* on lettuce produced in the Yuma region. Using the data analysis software, MAXQDA, we assessed the most commonly used terms and generated a word cloud from the gathered articles to visualize our findings (Figure 7). Thirty-four out of the 73 articles portrayed the benefits of fermented foods and drinks. However, even articles describing beneficial bacteria also presented a skepticism about their safety, with negative connotations. While food safety is of public interest, with regular reports of some new and increasingly alarming hazards in our food supply (Nestle 2003), conversations about the benefits of foods produced through small-scale partnership with microbes remain limited.

---

7  https://www.foodsafetynews.com accessed 15 July 2020.

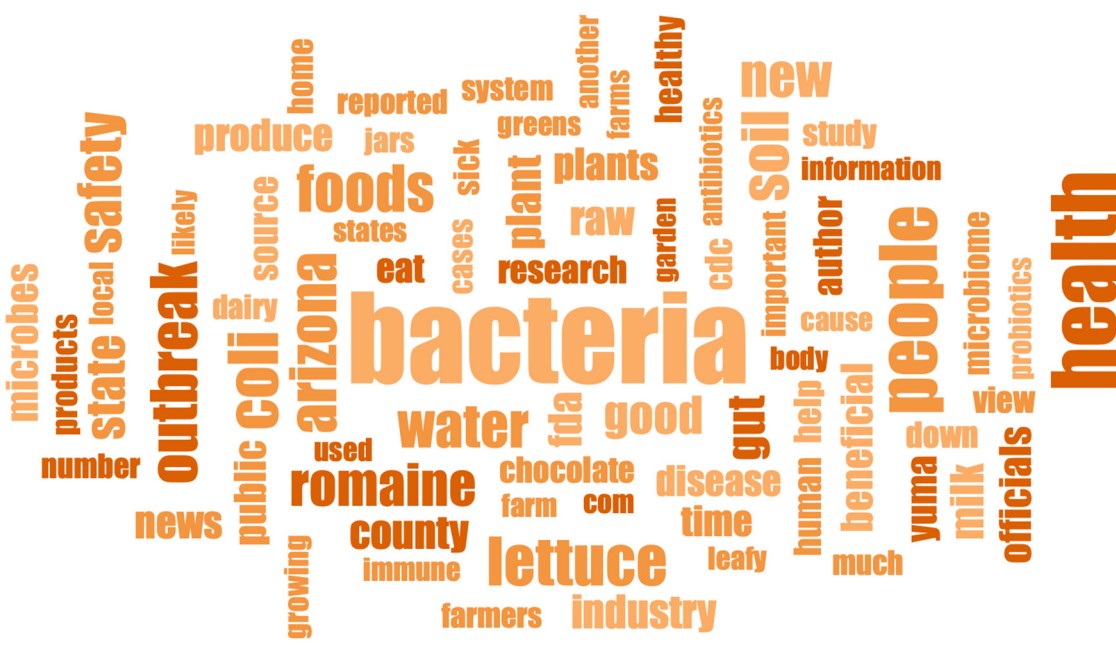

**Figure 7.** Word cloud created on MAXQDA showing the most recurring words from articles gathered from Arizona News from 2011 to 2019.

The word cloud highlights the significance that the romaine lettuce outbreak had on the news cycle. Microbial life's negative impacts are at the heart of the food safety lexicon, with just a few terms such as *beneficial* and *good* appearing primarily to illustrate the importance of soil microbes. This overall negativity overlooks the role many microbes such as *lactobacilli* play in keeping foods safe. This language carries consequences. It is reflected in the reactions of the leafy green industry in California and Arizona, where strong regulatory structures have been put in place through the Leafy Greens Marketing Agreement (Baur 2016), which requires preventative measures to ensure the safety of these greens. As these regulatory structures are implementable at scale, however, they target safety measures used by big industrial players rather than by small-scale and traditional food producers. A small subset of scholars has recently argued that food safety should be tailored to different scales and made adaptable to other modes of production (Hassanein 2011), but such ideas have not yet gained widespread traction. Issues of food safety continue to follow what Baur (2016) calls a "patchwork with big holes". We see the language in the informal food safety lexicon as one of those big holes, missing the voices of different human and microbial players who do not easily travel already-present regulatory or technological pathways.

*5.3. Making Gundruk, Safely*

Traditional Nepali practices for producing *gundruk* do have a form of ordering, albeit one rooted in traditional values and networks of trust in *gotho*, rather than in Western scientific practices. In order to produce *gundruk*, the women interviewed said that *gundruk* makers should wash their hands regularly, pull their hair back, and exclude animals from the production quarters. Interviewees added a temporal restriction to the hygienic practices mentioned above, pointing out that when women or girls menstruate, they do not participate in food preparation. These cultural guidelines act like food safety standards. They limit how food is prepared and by whom but rely on trust, tradition, and long experience rather than on regulations. These too constitute regulatory practices. The multiple steps in the process of producing *gundruk* ensure that only beneficial bacteria, such as lactic acid microbes, are introduced. Those steps include: (1) washing the leaves, covering them with a light mesh, then leaving them to wilt in the sun in order to avoid rot; (2) cutting leaves into small pieces to increase surface area for fermentation and squeezing

out any excess liquid; (3) leaving the leaves in an airtight container for a few days to a few weeks to complete anaerobic fermentation; (4) testing halfway through that period to determine whether the fermentation is proceeding correctly; and (5) sun-drying the fermented product covered with a light mesh to remove any remaining liquid. Finally, the *gundruk* is stored in jars for up to one year. These steps represent a holistic, rather than binary, system developed over generations and based on experience to ensure the flavor and safety of the product.

*5.4. New Lexicon, New Culture*

To the Nepali community *gundruk* represents a longing for home. Social justice scholars argue for understanding food as translocal, where traditional foods of immigrant communities are invited to become part of a new country's food system (Huang 2020). Currently, there is no avenue for *gundruk* with its accompanying socio-cultural aspects to enter U.S. marketplaces. It is a food produced within a regulatory framework that understands food safety from an alternative perspective. For *gundruk* to make the jump into the U.S. market would require a more holistic approach to food safety, one where microbes are understood not just as contaminating threats but also as companions and collaborators in making food safe. Research shows that live active bacteria, such as lactic acid bacteria (including *Lactobacillus*, *Streptococcus*, and *Leuconostoc*), provide health benefits that are shared with many other probiotic organisms (Tamang et al. 2018). Many traditional practices such as making *gundruk* have benefitted from these properties.

One step that could make room for traditional practices within current regulatory structures like those in the U.S., is to start by addressing the common narrative employed around microbes. Microbes are naturally ubiquitous across ecosystems and organisms (Dunn 2018), meaning that while some are harmful, many are beneficial. Yet, current regulatory language fails to address the healthful potentials of microbial life, as does the predominant public narrative around food safety. We anticipate that as public-facing narratives around the healthfulness of microbial life expand in conjunction with increased interest in the health possibilities of harnessing microbial potential, regulatory language may eventually follow. We anticipate that regulatory language could, rather, lead the way. This could happen through additional funding to spark research into tools capable of harnessing beneficial microbial life to promote food safety, such as new tools and techniques for identifying beneficial microbes found in fermented foods, as well as for the assessment of their presence and impact (Bokulich et al. 2016). Currently most tools available primarily assess harmful *E.coli*, *salmonella* and *listeria*. This change could also happen through funding research into traditional food production practices. As illustrated by the concept of *gotho* employed by Nepalese women, safety rules shaping traditional production already exist. Tacit knowledge, passed down through generations, can work in concert with contemporary food safety concerns; doing so could produce a more inclusive food system, but requires expanding conceptions of how we talk about and enact food safety.

## 6. Designing Food Artifacts for Social Justice and Sustainability
*6.1. Micro and Macro Level Food Systems by Design*

Imagine a new snack product based on corn. The cheap, tasty snack is marketed for children (usually identified as a consumer), easy to store, and profitable. The product is an efficient technical and economic solution in terms of production, transportation, delivery, and profit, even though it may be unhealthy (highly processed, high in added sugars) and unsustainable (high in chemicals and waste). At a micro level of design, packaging and advertising designers propose labels and campaigns that are appealing to parents using descriptors such as low-fat, vegan, vitamin-loaded, or convenient. Although products like this imagined corn snack are not the sole cause of food injustice and health disparities, it, along with similar products, contributes to complex problems, such as childhood obesity and unsustainable industrialized agricultural practices. While the designers of this product seek to improve profitability, the product generates other consequences. It's presence, and

designed deliciousness, may exacerbate systemic racism and health deficiencies, especially for vulnerable or disadvantaged populations.

Next, consider this corn snack from a macro design level. A local policymaker might propose improvements to current school lunch standards. Usually, policy makers follow national nutrition guidelines and consider economic viability, but they are also subject to lobbying from national industries or local businesses. As policymakers seek to optimize technical issues around food production, transportation, preparation, and economic dynamics, they may pay limited attention to specific social and health needs of the community or the environmental sustainability of their program, ignoring issues such as packaging waste. Even when they follow national guidelines, policymakers may not address dietary imbalances or cultural needs among minorities. They will always encounter highly processed food products that are high in added sugars but fewer all-vegetable or high protein options. Given the structural disadvantages faced by students who depend on school food (see Section 1) poor food choices by school administrators can be doubly harmful, perpetuating structural food inequalities limiting what foods they find at home. Furthermore, poorly designed school menus, which reflects mainstream food preferences or cultural biases, such as gender stereotypes or Mediterranean food imaginaries, may push cultural minorities away from healthier traditional alternatives.

### 6.2. Critical and Systemic Design

In previous sections of this article, authors have discussed various social structures that privilege dominant cultural practices and generate gendered and racial injustices. The examples of micro- and macro-level designed food artifacts highlighted in this section also illustrate how decision-makers in food systems produce injustices. While there may not always be clear individual wrongdoers in these situations, food systems have been designed via accumulated human decisions. Thus, there is an urgent need for a critical and systemic approach when designing what can be considered food artifacts.

Food products, food services, and food systems can be conceptualized as designed artifacts because they emerge from design activities at many levels: from the micro level of designing consumer products to the macro level of designing organizations and policies central to national and international food systems. The term *design* is appropriate because it means changing any artificial situation into a preferred one (Simon 1969, p. 111). In the domain of food production and consumption, product designers propose food products, service designers propose food delivery systems, school administrators design food menus, and policymakers design food guidelines for their communities. All of these designers aim to achieve preferred situations, and they may have success in their own terms. However, their preferred situations are often driven by economic or technical goals. Like product designers in general (i.e., industrial designers, packaging designers, and/or advertising designers), they may focus on elements such as visual appeal, product transportation and preservation, or consumer preference data, while passively adopting cultural biases such as gender stereotypes or popular dietary trends like the Mediterranean diet. Policymakers/designers will also focus on logistics, ease of implementation, guideline compliance, affordability, or resident majority preferences. These pushes and pulls narrow design practices, focusing on short-term results driven by capital. As a result, food-related design exacerbates already-existing injustices and undercuts environmental sustainability goals. To counter such tendencies, design practices offer *critical* and *systemic* approaches for creating socially just and environmentally sustainable food systems.

First, a critical approach expands designers' work from crafts to social action and ethical practice. Critical design, also known as speculative design, aims to question social norms and the social status quo. Speculative design offers creative alternatives and reactions to design consumption and efficiency. Speculative design artifacts are conceptual provocations intended to generate social debates (Dunne and Raby 2013). These conceptual artifacts usually work at a micro level with physical products that create provocations without specific practical goals. Some speculative designers argue that designers can no

longer afford an apolitical practice, which means designers question the rich and powerful, whose decisions are guided primarily by economic growth. They charge designers to understand that technical solutions are insufficient and to adopt a post-growth political position. Designers should challenge capitalism by proposing bold collective action for environmental sustainability and social justice. Even without the leverage, agency, or power to effect wide scale social change, designers can start with awareness, develop a critical reflective practice, and develop alternative tools that address social justice and sustainability factors present in any design situation (Nardi 2019).

Second, systemic design approaches provide frameworks for design action through sense-making of complexity and future-oriented participatory activities. This is particularly relevant for food artifacts that depend on complex interests and tensions that result in long term effects for communities. One emergent systems-oriented discipline is 'transition design,' which aims to generate long-term visions of sustainable futures while challenging existing socio-economic and political structures (Irwin 2015). Terry Irwin's Transition Design Framework (Irwin 2018) challenges designers to perform activities organically by visualizing the system that will be addressed, understanding cultural and historical context, involving multiple stakeholders, co-creating desired future visions, designing interventions, and observing long-term results.

Irwin's approach is promising for designing food artifacts. Food problems at micro and macro levels require boldly challenging existing social structures while involving stakeholders. For example, a designer of food policies, should actively explore ways to restore the rights of minorities with actions, such as giving voice to the community, and assess continuously how decisions are harming or benefiting them.

Another systems-oriented discipline is DesignX (or design for complex sociotechnical problems). DesignX highlights the psychological, social, political, economic, and technical challenges of macro level systems (Norman and Stappers 2015). For example, redesigning a school food menu is not just a problem of selecting food items; rather, it is a challenge involving political-power interactions between communities, food industry interests, and government officials. The problem also involves issues of transportation, storage, economy, nutrition, and health. At this level designers should focus on long-term collaborative implementation and 'muddling through' the situation by proposing modular-incremental solutions. Similar to transition design, DesignX centers on long-term action. However, DesignX suggests a technical approach to cope with political tensions, instead of challenging them. While these emergent design approaches—transition design and DesignX—seem to center the discourse at macro levels, designers at all levels should adopt mindsets for more critical and systemic practices.

### 6.3. A Critical and Systemic Approach to Designing Food Artifacts

Critical and systemic design approaches must address issues of gender and racial injustices in food systems by being critical and systemic at all levels, from micro to macro. A critical and systemic approach to design requires understanding and anticipating how design activities at micro and macro levels might result in unintended food injustices and how they could instead promote justice and sustainability at all levels. Beyond practical models, this entails intensive efforts and strong commitment to prevent (re)producing social injustices and undesired environmental harms. The following principles of a critical and systemic approach to design can be used by designers of food artifacts at all levels (see Figure 8).

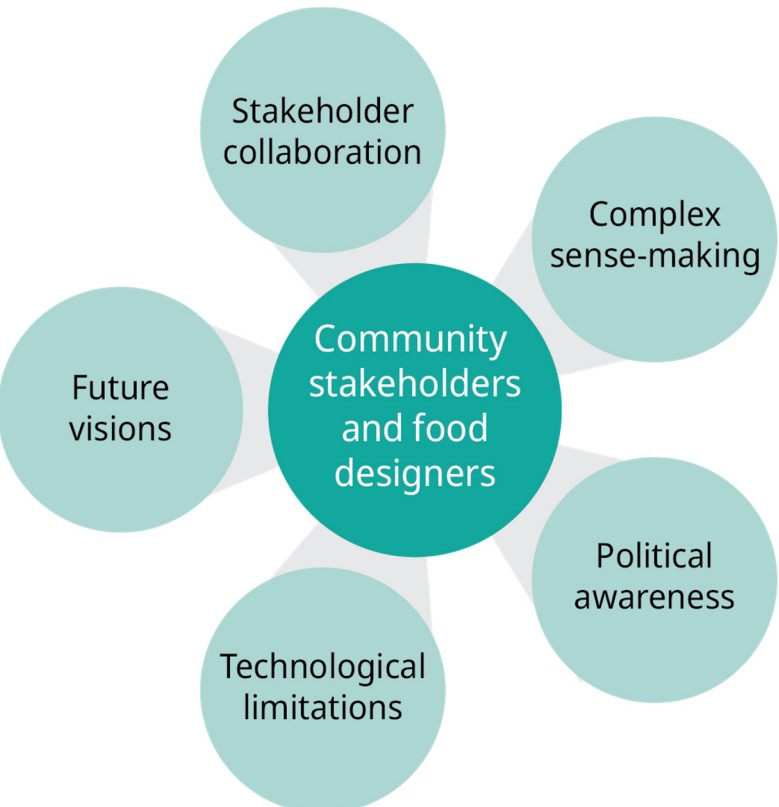

**Figure 8.** Principles for a critical and systemic approach to the design of food systems.

First, food designers should facilitate stakeholder collaboration. Usually food system designers act as experts who make decisions for others. Some argue that they use empathy to design better proposals for people. Yet, this approach is insufficient because it maintains designers' expert roles. Members of communities such as parents, teachers, or minority leaders should not only have a voice but also an ability to shape their own food systems. Designers should promote and facilitate such involvement.

Second, food designers should focus on complex sense-making of the food and social systems. Food designers' work is set up to solve practical and economic challenges. Understanding broad social interactions in context should also be part of the design process. This is achieved when there is more time for making sense of complex situations, reframing problems, and making visible their complexity, along with the ability to challenge power forces in the food systems.

Third, food designers should have political awareness and assume an ethical practice. Even efforts to neglect politics is a political choice. Designers of food artifacts should increase awareness of political implications—or situational power relations—and make ethical decisions. From the typeface choice in a food snack package to a school food menu guideline, designers' decisions entail political framing and consequences. For example, including a picture of a male cooking on food labels or adopting Indigenous foods in school lunches can be seen as a political design action directed toward reinforcing gender and racial equality. Thus, community participation is not just a tool to legitimize hidden political agendas, but a bold action to give agency to community members in shaping their food policies.

Fourth, food designers should recognize technological limitations as they address design situations. In today's world, many people imagine that technology will solve any problem. For example, industrial agriculture has been marketed as a way to feed billions, without attention to its environmental hazards and cultural harms. While it is tempting to believe that industrial technologies, technical guidelines, artificial intelligence inventions,

or persuasive technologies are the key to a future food system, food designers should also prioritize lifestyle possibilities, ethics of care, and respect for humans and non-humans.

Finally, food designers should work with stakeholders to create future visions that prevent the unintended risks and benefits of their practices. These visions should promote social justice and environmental sustainability. Future-oriented practices can also reveal and mitigate the limits of science through creative speculation. While no crystal ball can prevent all unintended consequences, trying out alternative scenarios can help. In considering alternative plans, designers should follow up, observe patiently, propose incremental implementation activities, and adjust their proposals for the long term.

These five principles are not a comprehensive list for a critical and systemic approach to food design. Their purpose is to present alternative practices that could help reduce the social injustices and environmental unsustainability of mainstream decisions about food artifacts. In the concluding segment of this article, authors will briefly discuss *critical* and *systemic* approaches to design and apply essential principles for designing socially just and environmentally sustainable food systems.

### 7. Conclusions: Using Critical and Systemic Design Principles to Promote Food Justice and Sustainability

As the authors of this article investigated approaches for turning our analyses into action, we decided to adopt principles of critical and systemic design (Section 6) as a framework. Recognizing that all aspects of food systems and cultures are designed allowed us to think about ways in which they could be redesigned. Our goal is two-fold: addressing food injustices faced by marginalized groups in American society; and advancing environmentally sustainable food production and consumption practices. Our conclusions about reaching those dual goals parallel people-centered work by sustainability scholars who recommend increased community participation, attention to equity and justice, redistribution of power, and implementation of democratic processes (Nicol and Taherzadeh 2020).

We seek not to provide definitive answers or solutions to the injustices and unsustainable practices our research reveals. Rather, we hope to identify a process or series of steps for developing actionable strategies applicable across varied circumstances, such as those in our article: the background social structures curtailing Dolores's access to healthful food for her family; Joselyn's unhealthy relationship to the food she desires and consumes; the underappreciation of Indigenous foodways in the U.S.; and food safety regulations that unfairly target immigrant food products and manufacturing processes.

One critical design principle we recommend adopting is stakeholder collaboration, which could be used to facilitate equitable decision-making in developing food products and systems. Too often decisions about issues surrounding school lunches, for example, are made by agencies and institutions without much public engagement. Thus, those decisions may protect corporate interests more than human health and well-being. They may also inadequately consider what constitutes a culturally appropriate, environmentally responsible school lunch in a particular community. By the same token, U.S. farm policies made without consulting diverse publics may inadequately regulate the use of pesticides, synthetic fertilizers, and the conditions of farm laborers.

Stakeholder collaboration can produce different kinds of decisions. For example, requiring community membership on zoning and regulatory boards that can incentivize grocery stores to locate in neighborhoods where there are currently none ("food apartheid") can give agency to community residents in the design of their food systems. Likewise, new, more robust governance approaches that include the entire spectrum of food producers, large and small, could result in expanded definitions of food safety. Such inclusion could, for example, enable the Nepali women producing *gundruk* in Arizona to challenge biases in the current legislative environment, which is accustomed to considering larger-scale, industrialized food production. This outcome would require going beyond the current comment period approach in U.S. rule-making, as well as finding a way to maintain best practices from Western science while also making space for additional practices.

Such changes could, in turn, expand democratic participation and increase marginalized communities' sense of belonging.

The principle of expanding stakeholders includes promoting processes of collaboration on future visions. Bringing together diverse stakeholders can be challenging, but techniques such as back-casting from some agreed upon goals to the steps required to reach them and scenario design for discovering both conflicts and overlaps can be useful tools. Collaboration in constructing future-oriented agricultural policies using such tools could, for example, simultaneously address gendered and race-based food injustice and improve the sustainability of the American food supply. A future vision of gender justice could emphasize how including and supporting more women farmers could expand sustainable agricultural practices, as women tend to advance such practices more than men farmers do (Barbercheck et al. 2014; Sachs et al. 2016; Wright and Annes 2020; Costa 2010; Harper 2020; de Boer and Aiking 2017). Women farmers also tend to adopt practices that reduce the impact of climate change, preserve heritage seeds, and regenerate the soil (Gohal 2017). Greater gender and racial justice in farming and food production could also help to revive lost sustainable food cultures, like *gundruk*, among immigrants and Indigenous Americans. Those diverse food cultures are often dominated by women and approach food production with a sense of reverence for natural resources and with restraint in their use (Klindienst 2006).

Stakeholder engagement and future visioning cannot on their own address cultural issues, however. The many gendered, sexualized, and racialized prescriptions for food preparation and habits of consumption that have characterized U.S. foodways are embedded in preconceptions about racial differences, masculinity and femininity, and appropriate gender roles. They do not necessarily result from regulations or policies. Rather, they are historically contingent reflections of gender and racial politics, which we understand as culturally driven power relationships. In the U.S., this includes histories of settler colonialism, patriarchy, and misogyny, and even appear in some non-traditional or marginalized cultural arenas. Food producers and marketers often reinforce those attitudes and beliefs and ignore their histories, as well as the profit motive that dominates the American economic system and recreates these unequal power relationships. Producers and marketers may even claim that they are only meeting public demand and have no responsibility for the individual and cultural consequences of the desires they create.

Utilizing the design principles of complex sense-making and political awareness is the best way to address deep cultural assumptions and stereotypes. Reducing the implicit racialized and sexualized gender injustices inherent in current food practices and cultures in the U.S. requires understanding and making visible the complexity of gender, sexual, and racial ideologies, recognizing how the politics of White male dominance have shaped food choices and dietary practices, including eating disorders, and challenging the power dynamics in the American food system. Together these interconnected approaches could reduce harmful gender associations with foods, such as masculinity's association with meat and processed foods, and promote healthier and better balanced eating behaviors. Reducing gendered food associations could also contribute to more sustainable farming and food production practices (Counihan and Kaplan 1998). For example, disrupting the masculinized meat-heavy American food environment could make environmentally friendly plant-based diets more popular (Schösler et al. 2015).

In addition, developing political awareness of ethnic, racial, and gender hierarchies in the design of food policies could increase respect for cultural and biological diversity in American foodways. For example, national guidelines for school meals could create incentives for programs that provide children time and space to share and explore the cultural and social practices connected to different food traditions within their communities. By emphasizing the quality, origins, freshness, and taste of foods, school meals could increase awareness of the connection between the environment and food and encourage future generations to learn more about the Indigenous agricultural practices, food traditions, and peoples where they live.

We invite policy makers to consider possibilities for re-territorializing local and traditional foods, as well as the foodways of immigrant communities, which are often the province of women, back into their food systems. These changes would create a larger role in educational programs for food producers from non-European food traditions. Maria Parra Cano's culturally appropriate and successful work as a Xicana Indigena chef and educator who nourished herself, her family, and other members of her community back to health in Arizona is an excellent example of a food activist who recognizes and questions the political assumptions that promote a European model, such as the Mediterranean diet, as a universal solution to nutritionally based chronic diseases.

Finally, the American food system in the past 100 years has emphasized technical solutions and efficiency. Technical solutions can be valuable. For example, new technologies could offer spaces of opportunity for expanding the food production environment to include foods like *gundruk*. Existing "standards tell us what is relevant, what is valued, what is important; . . . by implication, they tell us what is not important" (Busch 2011). Current food safety standards situate microbial diversity in food systems as secondary to a narrow definition of food safety. Drawing on the design principles discussed here, U.S. Department of Agriculture policies could increase funding for research into methods that detect, evaluate, and promote the presence of beneficial microbes. Adjusting the food safety regime to allow diverse cultural groups to contribute to scientific definitions and the ensuing formal and informal lexicons shaping what is considered safe could be a small but critical step in addressing unintended injustices imposed by current approaches to managing food safety.

Too often, however, technological solutions entail reductionist and top-down approaches that struggle to account for the complexity of food systems. Thus, the U.S. food system suffers from technological limitations, such as an exaggerated focus on efficiency in food production. That exaggerated focus excludes important ethical and moral values that food systems should preserve, especially ensuring human well-being and planetary health. Recognizing such limitations of technological solutions—another critical design principle— as well as the socially and environmentally disruptive potential of food technologies, designers and policymakers should adopt solutions that advance those values.

We recognize that reaching solutions in complex systems is neither easy nor straightforward. Values may conflict. Economic realities may stymie progress. The goals of food justice cannot be reduced to a single principle, such as place-based food traditions, when other values—such as translocality, environmentally sustainable agricultural practices and/or racial and gender justice—may or may not coincide with that principle. The multi-phase process we recommend entails both honoring and critiquing traditions. Like democracy itself, achieving food justice is an on-going struggle and demanding balancing act. We hope that our design-based implementation tools prove useful for resolving various food-related challenges and for pursuing the intertwined goals of food justice and environmental sustainability.

**Author Contributions:** S.K., J.M. and J.V. contributed to the abstract and Section 1, introduction. They also authored individual sections (G.M.M., Section 2; S.K., Section 3, J.V., Section 4). C.S. and S.E.-S. wrote Section 5. G.M.M. wrote Section 6. All authors contributed to Section 7, the conclusion and edited the article as a whole. All authors have read and agreed to the published version of the manuscript.

**Funding:** Spackman and El-Sayed are grateful to the Swette Center for Sustainable Food Systems for funding that supported interviews and transcription.

**Institutional Review Board Statement:** Spackman and El-Sayed's study was granted exempt status by the Institutional Review Board of Arizona State University on 4 December 2019.

**Informed Consent Statement:** Informed consent was obtained from all subjects involved in the study.

**Acknowledgments:** The authors are grateful to Meredith Clark for her assistance.

**Conflicts of Interest:** The authors declare no conflict of interest.

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
