# Peer review of "Gendered and Racial Injustices in American Food Systems and Cultures"

_humanities, doi:10.3390/h10020066_

Round 1

Reviewer 1 Report

In my opinion the paper has a good idea and is a review. The methodology could be improved and also the implications, but at least a better consideration of a more recent and innovative literature. For instance, linked to the business activity in food sistem and consumers and methods, you can consider Boccia and Sarnacchiaro (2018) in the journal Corporate Social Responsibility and Environmental Management, 25 (2).. 

Author Response

We thank reviewer 1 for thinking the article is  a good idea and review. We have addressed some lack of clarity in methodology in the introduction (see lines 75-89) and have included a number of supporting and more recent citations in the introduction. 

Reviewer 2 Report

This article has the potential to make a valuable and pertinent contribution to the food justice literatures.

Overall, this article would be strengthened by 1.strengthening claims with evidence and reference to relevant literatures (see detailed comments below); 2. inclusion of a true case rather than fictitious character in Section 2, and; 3.a focussed edit by one of the authors so that the article has more coherence as a single article even if it does include a series of diverse cases by different authors.

Introduction

The Introduction reads really well although I wonder if it would be more clearly if lines 52-58 the sentences make it clearer that they are referring to:1.farming (52-56), and 2.the restaurant and food industry (56-58). I also think more needs to be made of the existing food justice literatures eg Alkon, Sbicca,

  • Alkon, A.H. Food Justice and the Challenge to Neoliberalism. Gastronomica 2014, 14, 27–40.
  • Alkon, A.H.; Mares, T.M. Food Sovereignty in US Food Movements: Radical Visions and Neoliberal Constraints. Agric. Hum. Values 2012, 29, 347–359.
  • Agyeman, J.; Giacalone, S. The Immigrant-Food Nexus: Borders, Labor, and Identity in North America; MIT Press:Cambridge, MA, USA, 2020.
  • Cadieux, K.V.; Slocum, R. What Does It Mean to Do Food Justice? J. Political Ecol. 2015, 22, 1–26.
  • Sbicca, Joshua. "Growing food justice by planting an anti-oppression foundation: opportunities and obstacles for a budding social movement." Agriculture and Human Values 29, no. 4 (2012): 455-466.
  • Slocum, R.; Cadieux, K.V. Notes on the Practice of Food Justice in the U.S.: Understanding and Confronting Trauma and Inequity. J. Political Ecol. 2015, 22, 27–52.

I also think if design is the core focus of article, it needs to be made more of in the Introduction and Abstract (see lines 904-906). For example design is neither mentioned in the abstract nor final paragraph of the introduction yet seems central to the Discussion and Conclusion and is mentioned as a keyword. This needs to be addressed - but perhaps this article is of relevance not only to product designers. This needs to be made more explicit in Introduction.

I also wonder if it would be constructive to make more of your own personal identities and postitionalities in the Introduction? Are you coming from diverse backgrounds and experiences of foodways for example? Just 1-2 sentences would suffice but I think this could enrichen and enliven the article.

Sections

Overall the sections are compelling but they could do with one person to read through and give the article a coherent voice and style even if the sections are written by different authors. For example, section 2 feels a bit too conversational at least in parts outlined below. Also, the structuring of each section could do with being more aligned eg in 3.1. the author introduces section 3.2. but in 2.1. the author does not introduce 2.2.

Section 2

Section 2.1-2.2. would make for a more compelling read if it was based on a true vignette or at least a fusion of different true experiences – particularly since the other sections seem to be based on case studies. Part of food justice requires that we learn from those with lived experience and it feels almost disempowering to present a fictitious case. How can this support a call for change if it is ficitious? I encourage the authors to think more about how a more compelling case can be made of the case of Lisa and also think the case would benefit from more information eg what time she has to leave in morning and not just get home from work and what age her children are. If not, there needs to be a justification of why a fictitious character is used in this section and why it is helpful but I personally feel a case based on some truth is more indicative.

Section 2.2. could also benefit with some references to other research on redlining and links between redlining and food deserts (this would also help it balance with section 3). Beware of the use of the term food desert – it has received critique and has been replaced by some authors such as Penniman as food apartheid. Perhaps the authors can at least allude to the contentious nature of ‘food desert’ which could detract from the political ecological nature of lacking access to affordable, healthy, nutritious food.

See for example: Penniman, Leah, and Karen Washington. 2018. Farming while Black: Soul Fire Farm's practical guide to liberation on the land.

Section 3

Generally I think the tone needs to be a bit less conversational and more succinct

-see for example line 270,

-grammar check on lines 249-250 “that’s because”.

-lose “so” – (line 287)

Would be helpful to add “drawing upon Joselyn’s case, this section will…” (line 257).

Lines 287-288 is this a case from present day? It feels a bit dated and reductive – be good to insert some feminist standpoint theory in this section and also to acknowledge the rise of queer feminist theory and the LGBQT movement and non-binary choices. Not to just oppose white straight married families with black straight married families. Perhaps this is not what was intended but at present it feels a little oppositional and based upon notions of white civil partnerships/marriages from the 1950s

Line 311-315 need some figures to back this up rather than just being anecdotal.

Line 329-331 feels like it also needs a reference or some kind of evidence.

Could you give 1-2 examples of the masculine coding of farming (Line 335-337).

Not sure about permissions for the Nando’s,Burger King and McDonalds advert. The Editorial team can attend to that but I definitely think you need permission to publish an image from @Naythemua twitter account (Figure 5).

Section 4

Section 4.2. makes a good point about the perpetuating of colonialist hierarchies of traditional diets through promotion of the MD. However, the claim in lines 477-481 needs to be checked the Willett report does reference other studies on Mediterranean diet. Eg.

  • Estruch R, Ros E, Salas-Salvadó J, et al. Primary prevention of cardiovascular disease with a Mediterranean diet. N Engl J Med 2013; 368: 1279–90.
  • de Lorgeril M, Renaud S, Mamelle N, et al. Mediterranean alpha-linolenic acid-rich diet in secondary prevention of coronary heart disease. Lancet 1994; 343: 1454–59. Henríquez Sánchez P, Ruano C, de Irala J, Ruiz-Canela M, Martínez-González MA, Sánchez-Villegas A. Adherence to the Mediterranean diet and quality of life in the SUN Project. Eur J Clin Nutr 2012; 66: 360–68.
  • Bhushan A, Fondell E, Ascherio A, Yuan C, Grodstein F, Willett W.
  • Adherence to Mediterranean diet and subjective cognitive function in men. Eur J Epidemiol 2018; 33: 223–34.

There could also be an option to make more of a connection to agroecological reterritorialisation eg in A. Wezel, H. Brives, M. Casagrande, C. Clément, A. Dufour & P. Vandenbroucke (2016) Agroecology territories: places for sustainable agricultural and food systems and biodiversity conservation, Agroecology and Sustainable Food Systems, 40:2, 132-144

Section 6

Who are you writing for? In Section 6, it seems as if you are speaking specifically to designers. Perhaps it would help the article to agree collectively who your key audience is and to make sure potential readers are not alientated when they read section 6 which seems very much directed to food designers. Are you also reaching out to community food activists, teachers, educators, farmers, growers, knowledge brokers, health workers, food industry representatives and other practitioners? Having a clearer idea of readership will help clarify voice and message. Could there also be a point here to make about how food research can also play an important role in this work?

Conclusion

There may be potential to link with the article by Wezel et al. (2016) outlined above regarding agroecological territorialisation and the need to connect not only with food justice but just sustainability? Could also be a link to be made with people-centred approaches to sustainable and just food systems (Nicol, P. & Taherzadeh, A. 2020. Working Co-operatively for Sustainable and Just Food System Transformation. Sustainability. 12(7), 2816.

Could you add a final paragraph with hope for future based on your proposal for place-based food traditions? What further work needs to be done to achieve these kinds of food systems? A few more sentences would help here. Instead of technological solutions what? One key challenge seems how to bring all these actors of diversity in an inclusive way to imagine future sustainable and just foodways.

I encourage the authors to attend to some of these questions in the hope that it will make a valuable contribution to the food justice literatures.

Author Response

Reviewer 2 - Overall comments

Please strengthen this article by 1.strengthening claims with evidence and reference to relevant literature; 2. inclusion of a true case rather than fictitious character in Section 2, and; 3.a focussed edit by one of the authors so that the article has more coherence as a single article even if it does include a series of diverse cases by different authors.

Response Reviewer 2:

Reviewer 2 said that a real case study would be better for section 2 or at least a fusion of real cases would be good. The author has chosen to fuse several cases, since the author has read dozens of accounts of women struggling to work and feed their children while living in areas, such as Oakland, CA, without good food options. Since the author does not have the consent of any of these women to publish their stories, s/he does not believe it is appropriate to publish someone’s story without consent. The author has however added details and references to scholarship about these situations. 

The article has been edited again by one of the authors for diction and grammar as well as for consistency in mode of presentation. However, we have made the choice to allow the different disciplinary perspectives, relationships to the study of food and justice, and transnational cultural backgrounds of the articles’ multiple authors to shine through. All the authors agreed to use principles from critical design as a unifying device for translating ideas into practice.  Thus,  we invite scholars, practitioners and other stakeholders to actively design food systems despite such differences.

Reviewer 2 - Introduction

The Introduction reads really well although please be more clear in lines 52-58 the sentences make it clearer that they are referring to:1.farming (52-56), and 2.the restaurant and food industry (56-58). Please add more food justice literatures e.g. Alkon, Sbicca, and highlight the design aspect of the article in the introduction and abstract. Please consider addressing 1-2 sentences on your personal identities and positionalities in the Introduction. Please include an introductory sentence or two at the beginning of each "piece" to help readers understand what the section is doing (e.g. 3.1. final paragraph). Please do a read through the article to give it one coherent voice and style 

Response Reviewer 2:

We have reworded the introduction and addressed issues of farming and relationships with the restaurant and food industry. We have supplemented the article with more current literature including Alkon, Sbicca. We addressed the question of one voice versus multiple voices and that we wanted to exemplify that what is needed to create a just sustainable food system is collaboration and interconnection without collapsing difference (as explained above).. We need to listen to different disciplinary approaches to these problems without trying to become a single discipline or voice. We did add a sentence to the abstract and the introduction that addressed the redesign of our food system as the solution to the problems discussed in the paper.  The Introduction clarifies the interdisciplinary nature of our work, but personal information cannot be included in a blind review. We have ensured that the different sections of the article tie into each other and have assigned one author to read through the entire manuscript to create more cohesion, although the primary reason for creating a multi-author (five separately authored sections) piece is to represent the different voices, disciplines, cultures and perspectives.

Reviewer 2 - Section 2

Please rewrite section 2.1-2.2. to either clearly indicate it is drawing on real life cases and/or draw on a real life case or a fusion of many cases.

Response Reviewer 2:

This case is drawn from the accounts of lived experiences of dozens of women trying to work and care for their children while living in areas that lack healthy food options. West Oakland is a particularly glaring example of a low-income minority neighborhood, surrounded by some of the most expensive real estate in the world, without access to full service grocery stores. The author did not  have consent to use a particular woman's story, instead decided to fill out the details of many women's stories, as it would be unethical to use the stories without the person’s consent.  

Reviewer 2 - Section 3

Please address the conversational aspect of section 3. Please revise paragraph in section 3.2 that begins "for white women" to better account for feminist standpoint theory/queer feminist theory/ etc. In addition, please include additional evidence/ citations to back up claims (lines 311-315; 329-331). Please  add examples of masculine coding. Please confirm permissions for image usage, especially from social media accounts.

Response Reviewer 2:

It's unclear what Reviewer 2 means by conversational tone; however, specific changes in language asked for in lines 249-50, 270, and 287 have been made. The author added Joselyn's case in original line 257. Joselyn's case is from a 2019 source, so contemporary (as original footnote indicated), although Jocelyn is an adult looking back on her childhood. The author has expanded all image captions to make their connection with the text clearer. Please note that all images are in the public domain, as the original footnote explains. The author has eliminated the tweeted image for Figure 5 and substituted an image of the same ad that has been published in numerous sources, all listed in the caption for that figure. The author made changes to paragraph between original lines 282 and 293 to better identify the cultural norms being discussed as heterosexual as well as racialized and to clarify how post-WWII expectations live on in the present. Added a subsequent paragraph featuring queer feminist theoretical approaches to stereotypes about lesbians and the food-giver role, as well as about lesbian mothers' relationship to expectations about children's diets. Cited several additional sources as well, including Alkon 2012, which Reviewer 2 recommended. Added reference to article's section 2 to point readers to data about structural inequities (original lines 311-15). Also added more examples of the masculine coding of farming (original lines 335-337, but now expanded). The author included data about the proportion of head chefs that are female (18.7 as of 2020) and of Michelin star recipients (9.2 percent as of 2019) as indicators of gendered differences in social and economic status in the food industry.

Reviewer 2 - Section 4

Please nuance claims about the references to the Mediterranean Diet in the 2019 Eat-Lancet Commission’s report on Food in the Anthropocene.

Response Reviewer 2:

The author edited this section to address the reviewer’s concern. Due to the focus of this article on racial and gender justice, the author did not add more references on the nutritional benefits of the Mediterranean Diet today.

Reviewer - Section 6

Please indicate who you are writing for in section 6, is it designers? Make sure you clarify who the key audience is so readers are not alienated. Please indicate if  you are reaching out to community food activists, teachers, educators, farmers, growers, knowledge brokers, health workers, food industry representatives and other practitioners? Possibly make a point about how food research can also play an important role in this work?

Response Reviewer 2:

The readers are humanists, humanities researchers, and practitioners from different fields that might act as (micro and macro) food designers in their roles. This section will help researchers understand ways to translate theoretical critique into actions. For practitioners, the section will offer key pointers for taking action.

Reviewer 2 - Conclusion

Consider linking the article by Wezel et al. (2016) regarding agroecological territorialisation and the need to connect not only with food justice but just sustainability? Could also be a link to people-centred approaches to sustainable and just food systems (Nicol, P. & Taherzadeh, A. 2020). Add a final  paragraph with hope for the future based on your proposal for place-based food traditions as well as what further work needs to be done to achieve these kinds of food systems? 

Response Review 2:

We added additional material from Reviewer 2’s recommended source (Nicol, P. & Taherzadeh, A. 2020) in order to  acknowledge the linkage between people-centered approaches to sustainability and the design approach to achieving just food systems. We think that Reviewer 2 may have misinterpreted our interest in foregrounding agroecological territorialization as the article’s primary goal. The first two paragraphs were substantially revised to clarify the purpose of the conclusion, and the final paragraph, which we added at the Reviewer’s suggestion, further explains the complex mixture of goals that we aspire to achieve through the solution-seeking process we recommend . We also added references to diversity among stakeholders in that part of the discussion, which was implied before but not explicit. The new final  paragraph demonstrates the complexities we see in the food justice issue and  emphasizes that we are offering tools not solutions. The issue of bringing diverse actors together to imagine future sustainable and just foodways is addressed earlier.

Reviewer 3 Report

The article presents several original points and compelling arguments. Some sections, specifically "introduction; n.2; n.3; n.6" need improvement in contextualization, scholarly reference and evidence, and a more flowing writing style.

Introduction: while it points out several important issues, it is not backed-up by any scholarly research, evidence with real examples, or recent studies.

2: using a fictional story, like the one of Lisa, can be a good idea but reading it felt and resulted to be incomplete and unrealistic. Also, perpetuated an image of women as victim and without "agency" (?) that this article is trying to argue against. Also, it is arguable that Lisa doesn't have agency... she can think and she did make her choices (she decided to... she wants...) - The "social structures" this section is referring to are very vague and don't offer specific examples for the reader to follow the main argument. What social structures? Is class anything related to social structure? Not once it has been mentioned, nor the economic factors as a source of a problem of access to food security. If these last 2 factors don't want to be part of this section, a better definition of social structures, and reference to scholarship on this topic, needs to be included. Also, the question of responsibility needs to be better contextualized. Overall, this sections needs improvement and more scholarly research needs to be included.

3: The story of Joselyn flows better and is better contextualized. Not sure though how a reference to a publicity of 1963 and a short paragraph on the period after WWII helps to make the argument. This section provides more scholarly research, yet I would recommend to review 3.2 and review the style, can we consider stereotypes as a form of lack of agency? Saying women do not have agency is taking away their power, which is also untrue as a matter of fact. An analysis of the images would add to this section and would help justify the choice of these images, instead of others.

Sections 4 and 5 are well written, flow, make good arguments and the scholarly research is well used. For section 5, review the use of "we". Who is this "we"? The last comment of 5.1 on microbial life reminded me of Anna Tsing and Donna Haraway and their works on mushrooms. Their scholarly work maybe an interesting add to this last statement.

Section 6: The language and style needs improvement. It is not clear how  "kids" pay attention to low-fat or vegan in the food labels... this can be easily argued against. Also this all section needs more recent research and a more flowing writing style. It is not clear how the paragraph of DesignX fits with the discourse of food justice.

Author Response

Reviewer 3 - Introduction

Please improve on the contextualization of the introduction sections 2,3 and 6, also add scholarly references and evidence and ensure a more flowing style. Please add scholarly research evidence, real examples and recent studies

Response Review 3:

We have improved the introduction and increased the number of scholarly references and evidence. We also ensured there is a better flow and stronger linkage to the design aspect of the paper.  

Reviewer 3 - Section 2

Please revisit Lisa's fictional story, so it doesn't perpetuate the image of women as victims, without agency. Specify the social structures, is class related to the social structure, possibly redefine social structures, as well as add reference to scholarship to this topic and responsibility needs to be contextualized.

Response Review 3:

We have addressed the problem of the fictional story based on reviewer 2’s suggestion and the author is now drawing from the accounts of lived experiences of dozens of women trying to work and care for their children while living in areas with limited food options. To clarify what the author is arguing: the social structures “Lisa” is caught in (race, gender, environmental circumstances and social ones) limit her agency-- that is why they are unjust. These social structures include gender norms, structural racism and overt racism resulting from a history of redlining, underfunded schools, and underinvestment in minority neighborhoods. These structures do not eliminate Lisa’s agency, but they curtail it.

Reviewer 3 - Section 3

Needs a better argument of the use of the publicity of 1963 and period after WWII. Please review section 3.2, can you consider stereotypes as a form of lack of agency? Saying women do not have agency is taking away their power, which is also untrue as a matter of fact. Consider an analysis of the images. 

Response Review 3:

The author made changes to paragraph between original lines 282 and 293 to better identify the cultural norms being discussed as heterosexual as well as racialized and to clarify how post-WWII expectations live on in the present.  The author also clarified the point that stereotyping reduces women’s agency and that being considered without agency is not the same as actually being without agency.  All image captions have been expanded to demonstrate their relationship to the text.

Reviewer 3 - Section 4 and 5

Are well written, but please address who is the "we" and consider adding content on Anna Tsing and Donna Haraway.

Response Review 3:

We appreciate your comments on the pieces being well written and flowing well. The "we" refers to the two authors that are co-writing this piece, but due to the blinded review their names were removed. We believe our names will appear at the heading of the section. We are indeed very influenced in our scholarly work by Tsing and Haraway, but given the scope of this paper and the limit to the number of words, we decided not to add more scholarly work to this piece.

Reviewer 3 - Section 6

Please work on improving and clarifying  "kids" pay attention to low-fat or vegan in the food labels. Also please add more recent research and improve writing style and connect the DesignX with the discourse of food justice.

Response Review 3:

The author clarified that it is kids' parents who are appealed to and edited the systems’ section and the designX paragraph with a better example related to food. The references used are all mostly recent, from 2010 onwards. 

Round 2

Reviewer 2 Report

This article is much improved and benefits from attention to sexuality and queer theories, reference to the authors' postitionalities as well as making a much clearer link with critical and systemic design principles and possibilities fo rmore sustainable and just food systems.

A few minor points...

The introduction is much improved with a pertinent and timely review of food injustice literatures.

Happy to see a reference to diverse positionalities of the authors.

The addition of the final paragraph of the Introduction clearly signposts readers to the contribution of the article.

Line 125 - maybe reference to Penniman here isn't necessary.]

Line 142 typo wrongs.

Line 146 - be careful here - is it individual or structural agency. Risks reading as the actions of individuals rather than products of systemic racism.

Section 2 is much improved but perhaps now a little long. Could a tight edit help synthesise the new additions?

Good to see inclusion of reference to sexualities and queer/feminist theories in Section 3.

Line 282 - could this just be food systems (instead of 'food cultures, systems, and industries')?

Line 406 - is penetration implied? Suggest rewording to somehow emphasise that oral sex rather than penetration may be implied!

Line 617 - would this also include cultural heritage?

Line 936 - space required - new paragraph for second point?

Line 1037 - also sexuality?

Line 1048 - 'the best' - or a useful framework/supports?

Line 1060 also sexual?

Author Response

Dear Editors and Reviewer,

Thank you for your careful reading and response. We have followed your suggestions; specific responses to points below are highlighted in the comments section of the document with track changes on.

Sincerely,

the authors

Reviewer 3 Report

The article has been edited following all peer reviewers recommendations and this second draft has significantly improved and is ready, in my opinion, for publication.

Author Response

To the reviewer,

Thank you for the time you took reviewing. We greatly appreciate your feedback on earlier versions.

Sincerely,

the authors